# Circular RNA CircFndc3b modulates cardiac repair after myocardial infarction via FUS/VEGF-A axis

Venkata Naga Srikanth Garikipati[1], Suresh Kumar Verma[2], Zhongjian Cheng [1], Dongming Liang[3], May M. Truongcao[1], Maria Cimini[1], Yujia Yue[1], Grace Huang[1], Chunlin Wang[1], Cindy Benedict[1], Yan Tang[1], Vandana Mallaredy[1], Jessica Ibetti[1], Laurel Grisanti[4], Sarah M. Schumacher[5], Erhe Gao[1], Sudarsan Rajan[1], Jeremy E. Wilusz [3], David Goukassian[6], Steven R. Houser[7], Walter J. Koch[1,8] & Raj Kishore[1,8]

Circular RNAs are generated from many protein-coding genes, but their role in cardiovascular health and disease states remains unknown. Here we report identification of circRNA transcripts that are differentially expressed in post myocardial infarction (MI) mouse hearts including circFndc3b which is significantly down-regulated in the post-MI hearts. Notably, the human circFndc3b ortholog is also significantly down-regulated in cardiac tissues of ischemic cardiomyopathy patients. Overexpression of circFndc3b in cardiac endothelial cells increases vascular endothelial growth factor-A expression and enhances their angiogenic activity and reduces cardiomyocytes and endothelial cell apoptosis. Adeno-associated virus 9 -mediated cardiac overexpression of circFndc3b in post-MI hearts reduces cardiomyocyte apoptosis, enhances neovascularization and improves left ventricular functions. Mechanistically, circFndc3b interacts with the RNA binding protein Fused in Sarcoma to regulate VEGF expression and signaling. These findings highlight a physiological role for circRNAs in cardiac repair and indicate that modulation of circFndc3b expression may represent a potential strategy to promote cardiac function and remodeling after MI.

[1] Center for Translational Medicine, Lewis Katz School of Medicine, Temple University, Philadelphia, PA 19140, USA. [2] Division of Cardiovascular Diseases, Department of Medicine, The University of Alabama at Birmingham, Birmingham, AL 35294, USA. [3] Department of Biochemistry and Biophysics, University of Pennsylvania Perelman School of Medicine, Philadelphia, PA 19104, USA. [4] Department of Biomedical Sciences, University of Missouri, Columbia, MO 65211, USA. [5] Lerner Research Institute, Cleveland Clinic, Cleveland, Ohio 44195, USA. [6] Zena & Michael A. Weiner Cardiovascular Institute, Icahn School of Medicine at Mount Sinai, New York, NY 10029, USA. [7] Cardiovascular Research Center and Department of Physiology, Lewis Katz School of Medicine, Temple University, Philadelphia, PA 19140, USA. [8] Department of Pharmacology, Lewis Katz School of Medicine, Temple University, Philadelphia, PA 19140, USA. Correspondence and requests for materials should be addressed to R.K. (email: raj.kishore@temple.edu)

Cardiovascular disease is the leading cause of death in the United States, accounting for nearly one in every seven deaths. Although classical pharmacologic treatment strategies have improved, cardiovascular outcome, survival, and prognosis of heart failure patients remain poor[1]. This highlights the need for further understanding the underlying mechanisms and developing innovative effective therapies for cardiovascular diseases.

Several potential therapeutic agents for post-MI repair have been investigated in recent years, including non-coding RNAs (ncRNAs)[2–4]. Circular RNAs (circRNAs) have recently been identified as a new class of regulatory RNAs with gene regulatory roles[5–8], however their role in cardiovascular injury and repair is not well elucidated. These covalently closed transcripts are generated when the pre-mRNA splicing machinery backsplices to join a downstream 5′ splice site to an upstream 3′ splice site (e.g. the end of exon 2 is joined to the beginning of exon 2). CircRNAs are naturally resistant to degradation by exonucleases and have long half-lives, but most are expressed at low levels. Nevertheless, some circRNAs are expressed at higher levels than their associated linear mRNAs and have proposed roles in cancer[9] and neurological diseases[10]. For example, some circRNAs sequester specific microRNAs (miRNAs)[11–16] or RNA-binding proteins (RBPs)[17,18], whereas others may be translated into proteins[19–21] or directly interact with regulatory proteins[22,23]. CircRNAs are additionally emerging as potential biomarkers of disease diagnosis and treatment,[24,25] including for heart disease[10,11]. Specifically, circRNAs derived from the Cdr1as and FOXO3 loci have been reported to have functional roles after cardiac injury[26,27] and the MICRA circRNA can predict LV dysfunction in human patients[28]. However, whether modulation of specific circRNAs in vivo can attenuate left ventricular dysfunction after experimental MI is not yet known. To address this question, we used circRNA profiling of cardiac tissue from post-MI mice and identified circFndc3b (derived from exons 2 and 3 of the Fndc3b gene) as a circular RNA significantly downregulated post-MI. The Fndc3b gene has been shown to be frequently upregulated by >30% in esophageal, lung, ovarian, and breast cancers and its protein product targets the PI3K/AKT pathway involved in survival signaling[29,30]. By modulating circFndc3b levels, we provide evidence that overexpression of circFndc3b in ischemic hearts can reduce cardiomyocyte apoptosis, enhance angiogenesis, and attenuate LV dysfunction post-MI in mice. Mechanistically, we provide evidence that circFndc3b enhances vascular endothelial growth factor-A (VEGF-A) expression and signaling via its interaction with the RNA binding protein fused in sarcoma (FUS).

## Results

**CircRNA profiling in MI hearts**. To identify changes in circRNA expression patterns post-MI, we performed circRNA microarray analysis using RNA isolated from sham or MI mouse hearts at day 3 post-MI. Of the 14,236 probes for mouse circRNAs present on the microarray, expression of 1723 circRNAs was detected. Of these, 82 circRNAs were consistently differentially expressed, including 41 up-regulated and 41 down-regulated in the post-MI hearts compared to sham hearts (Fig. 1a). Using divergent primers and RT-qPCR, we validated that several of these circRNAs were indeed differentially expressed, including the 215-nt circFndc3b transcript that was significantly down-regulated in the post-MI hearts compared to sham controls (Fig. 1b and Supplementary Fig. 1). Upon examining mouse hearts at different time point's post-MI, we further found that circFndc3b expression continued to decrease throughout the follow-up period of 6 weeks post-MI compared to sham hearts (Fig. 1c). In stark

contrast, the levels of linear mRNAs derived from the Fndc3b gene did not change in MI hearts compared to sham controls (Supplementary Fig. 2A). Next, we used cell fractionation to examine where circFndc3b is expressed in the post-MI hearts: circFndc3b was significantly down-regulated in endothelial cells and cardiomyocytes (Fig. 1d, e) but not in fibroblasts (Supplementary Fig. 2B). Interestingly, humans express an ortholog (86% identity) of this circRNA (Fig. 1f), and we found that expression of the human circFndc3b ortholog was also significantly reduced in the LV tissues of ischemic cardiomyopathy patients compared to non-failing hearts (Fig. 1g), unlike the linear Fndc3b mRNA (Supplementary Fig. 2C–D). These results suggested that circFndc3b, but not linear Fndc3b transcripts, may have clinical significance and be involved in the pathophysiology of MI.

**Over-expression of circFndc3b enhances endothelial cell function and reduces cardiomyocyte apoptosis**. To understand the role of circFndc3b in cardiovascular biology, we first mapped the backsplicing junctions of the mouse and human orthologs using Sanger sequencing after RNase R treatment (Supplementary Fig. 3). As expected, these endogenous circRNAs are generated from exons 2 and 3 of the Fndc3b gene and the intervening intron is removed from the mature transcript. We then generated circFndc3b overexpression plasmids (Fig. 2a) that were transiently transfected into mouse cardiac endothelial cells (MCECs). Note that short sequences were added to the circFndc3b exons near the back splicing junction, thereby enabling plasmid-derived circRNA to be distinguished from endogenous circFndc3b. RT-PCR confirmed that the plasmids generated mature circRNAs (Fig. 2b) with the appropriate splicing junctions (Supplementary Fig. 3B), with endogenous circFndc3b levels also increasing (Fig. 2c). Nevertheless, overexpression of circFndc3b had no effect on expression of the linear Fndc3b mRNA (Fig. 2d, e).

We then examined changes in angiogenic gene expression in MCECs after transfection with the circFndc3b overexpression plasmid or a control plasmid that does not generate a circular RNA. The RT² Profiler™ PCR mouse angiogenesis array (Qiagen) revealed multiple genes that were modulated with circFndc3b overexpression (Fig. 3a). Independent RT-PCR experiments validated that VEGF-A expression was upregulated in response to circFndc3b overexpression (Fig. 3b). Because VEGF-A is a potent cardio protective molecule[31], we assessed the role of circFndc3b on apoptosis of MCECs. Overexpression of circFndc3b significantly reduced apoptosis in MCECs exposed to 100 μM $H_2O_2$, as shown by decreased number of TUNEL positive cells compared to the control cells (Fig. 3c, d). VEGF is also well known to regulate angiogenesis[32] and we next assessed the functional role of circFndc3b in tube formation ability of endothelial cells. Overexpression of circFndc3b in HUVECs significantly enhanced tube formation ability compared to the control plasmid (Fig. 3e, f). These observations indicate that circFndc3b regulates endothelial cell function in vitro.

As apoptosis is an important mechanism in the loss of myocytes post-MI[33], we determined the in vitro effect of circFndc3b overexpression on the survival of primary cardiomyocytes and cardiomyocyte cell lines in response to stress. Overexpression of circFndc3b in H9c2 cells (Supplementary Fig. 4A) significantly reduced apoptosis when the cells were exposed to serum deprivation and hypoxia stress (1% $O_2$, 48 h) (Supplementary Fig. 4B–C). Likewise, circFndc3b overexpression reduced apoptosis in neonatal rat ventricular myocytes (NRVM) subjected to $H_2O_2$ (100 μM) stress for 2 h (Supplementary Fig. 4D–F). To have a translational perspective, we further tested the same hypothesis in human ventricular myocytes (AC16 cells). Overexpression of circFndc3b reduced apoptosis in AC16 cells

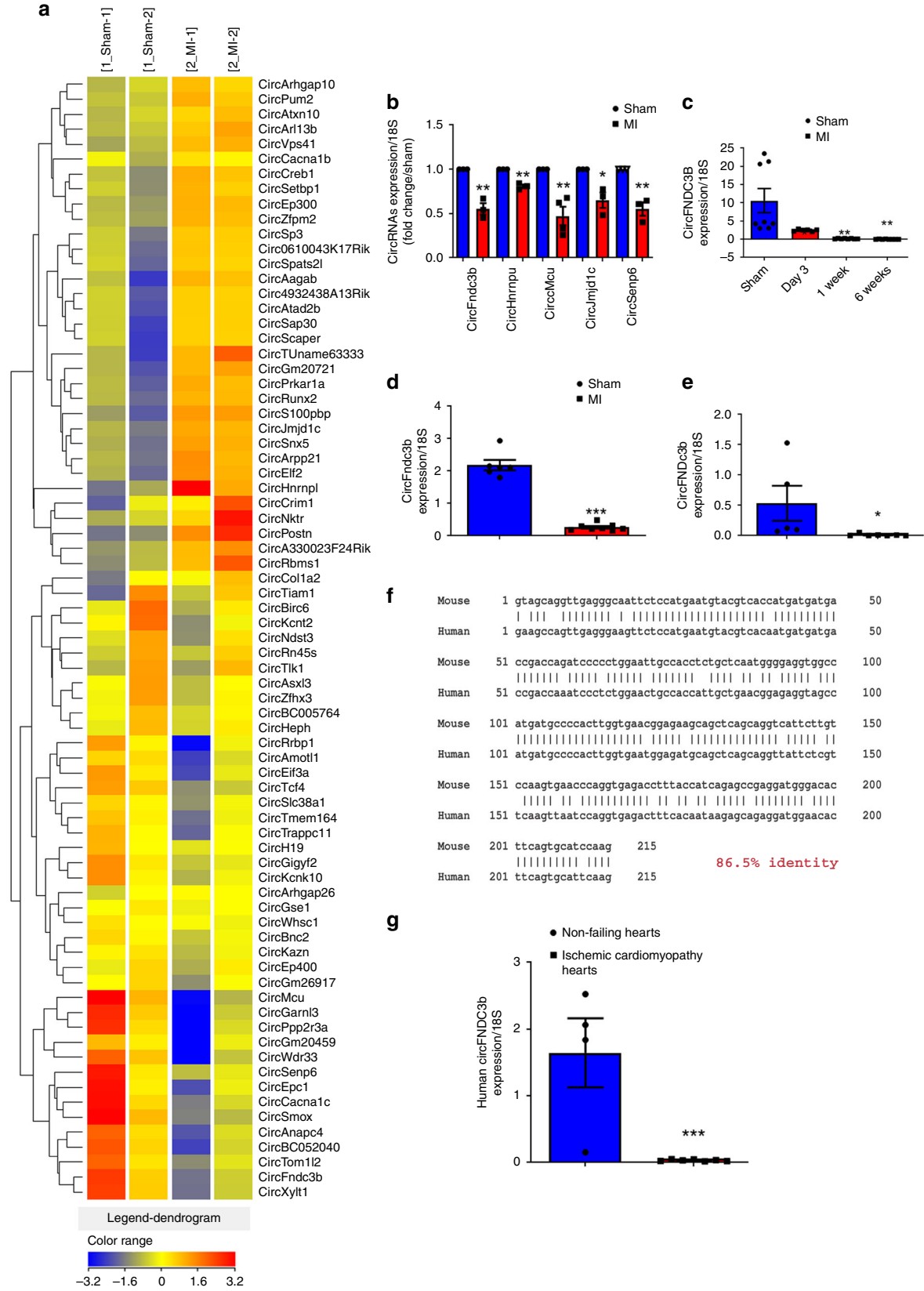

subjected to $H_2O_2$ (100 μM) stress for 24 h (Supplementary Fig. 4G–I). In total, these results demonstrate that circFndc3b overexpression generally reduces apoptosis in cardiomyocytes in vitro.

**CircFndc3b overexpression attenuates LV dysfunction post-MI in mice.** Reduced circFndc3b levels in mouse hearts post-MI (Fig. 1c) and in failing ischemic human hearts (Fig. 1g) suggested a possible physiological role for circFndc3b in MI

**Fig. 1** CircRNA profiling in sham and MI hearts: **a** Heat map of circRNAs differentially expressed at 3 days following left anterior descending coronary artery (LAD) artery ligation. [1_sham-1 and 1_sham-2] denotes expression in sham animals while [2_MI-1 and 2_MI-2] denotes animals subject to LAD ligation. Relative expression values are indicated on the color key and increase in value from blue to red ($n = 2$ mice/group); **b** Validation of differentially expressed circRNAs by RT-qPCR, normalized to 18S rRNA. $n = 3$ mice/group. Data are Mean ± SEM. *$p < 0.05$, **$p < 0.01$ vs sham hearts (two-sided unpaired students $t$-test); **c** RT-qPCR analysis of circFndc3b expression at different time points in post-MI LV tissues ($n = 5$ mice) compared to sham ($n = 8$ mice), normalized to 18S rRNA. Data are Mean ± SEM, *$p < 0.05$, **$p < 0.01$ vs sham hearts (one way Anova); **d**, **e** RT-qPCR analysis of circFndc3b in isolated endothelial cells ($n = 4$ mice /group) and cardiomyocytes ($n = 4$ mice in sham group and $n = 6$ mice in MI group) from 3 days post-MI LV tissue compared to sham controls, normalized to 18S rRNA. Data are Mean ± SEM. ***$p < 0.001$,*$p = 0.054$ vs sham hearts (two-sided unpaired students $t$-test). **f** Pairwise alignment of the human and murine circFndc3b sequences. **g** RT-qPCR analysis of the human ortholog of circFNDC3b in non-failing heart tissues ($n = 4$) and in heart tissues from patients with ischemic cardiomyopathy ($n = 7$), normalized to 18S rRNA. Data are Mean ± SEM. ***$p < 0.001$ vs non-failing human hearts (two-sided unpaired students $t$-test)

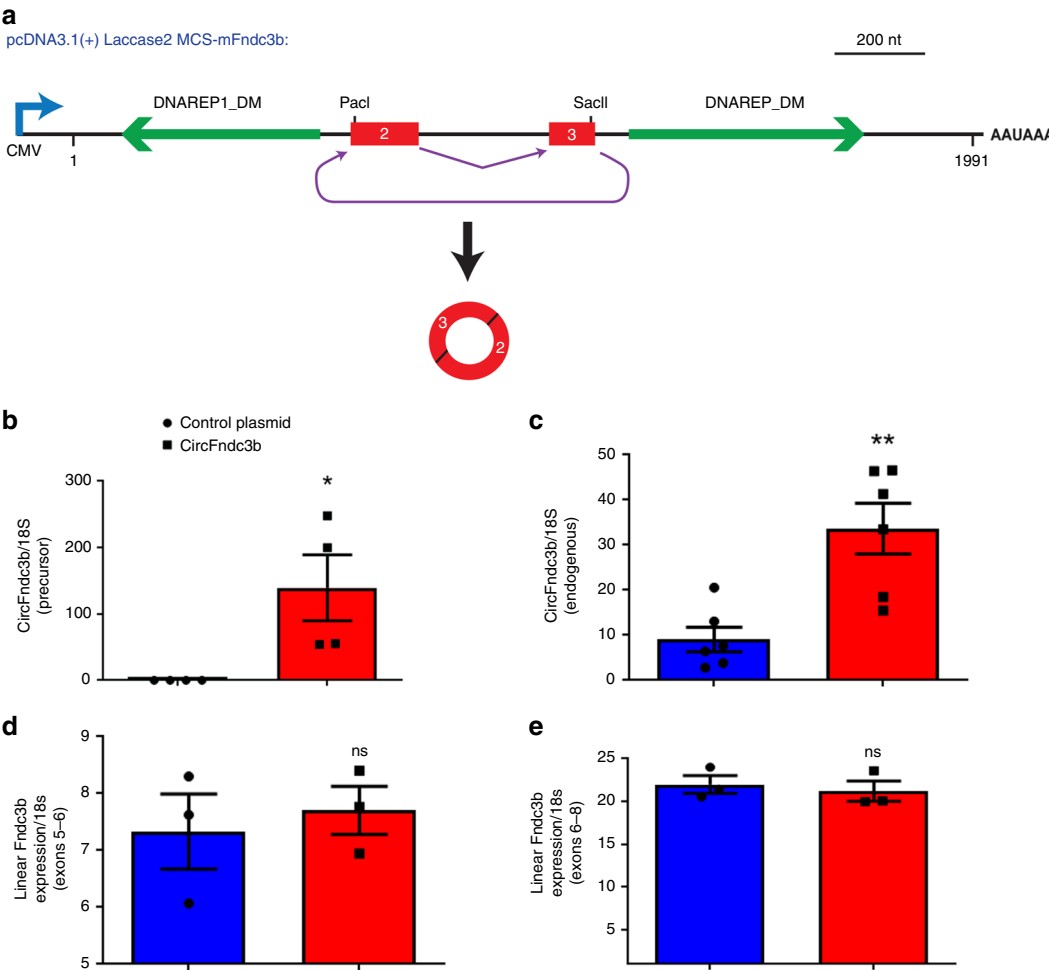

**Fig. 2** Generation of circFndc3b overexpression plasmids: **a** Exons 2 and 3 of the mouse Fndc3b gene were inserted between the laccase2 flanking introns, which harbor inverted DNAREP1_DM repeat sequences that promote back splicing. PacI and SacII sites were included within the Fndc3b exons to allow plasmid-derived circFndc3b to be distinguished from the endogenous circRNA. RT-qPCR analysis of precursor-circFNDC3b **b** and endogenous circFndc3b **c** expression in control plasmid or circFndc3b overexpression plasmid-treated mouse cardiac endothelial cells, normalized to 18S rRNA. Data are Mean ± SEM of four independent experiments/group. *$p < 0.05$, **$p < 0.01$ vs control plasmid (two-sided unpaired students $t$-test). **d**, **e** RT-qPCR analysis of linear Fndc3b mRNA levels in control plasmid or circFndc3b overexpression plasmid-treated cardiac endothelial cells, normalized to 18S rRNA. Data are Mean ± SEM of four independent experiments/group. Not significant (ns) vs control plasmid (two-sided unpaired student's $t$-test). The round dots represent control plasmid and square denotes circFndc3b plasmid

pathophysiology. Therefore, we tested if exogenous delivery of circFndc3b may attenuate LV remodeling and dysfunction post-MI in mice. We generated AAV9 viral particles that express circFndc3b under control of a CMV promoter (Fig. 4a) and intramyocardially injected $1 \times 10^{12}$ vp/ml AAV9 circFndc3b or an AAV9 control immediately post-LAD ligation. RT-PCR confirmed that the AAV9 vector generated mature circRNAs

(Fig. 4b) with the appropriate splicing junctions (Supplementary Fig. 3B). In addition, endogenous circFndc3b levels increased in the LV tissue compared to controls 8 weeks post-MI in mice (Fig. 4c). Like was observed in vitro (Fig. 2d, e), expression of the Fndc3b linear mRNA was unchanged upon circFndc3b over-expression (Fig. 4d, e). We then determined the effect on LV function by echocardiography in mice injected with either AAV9

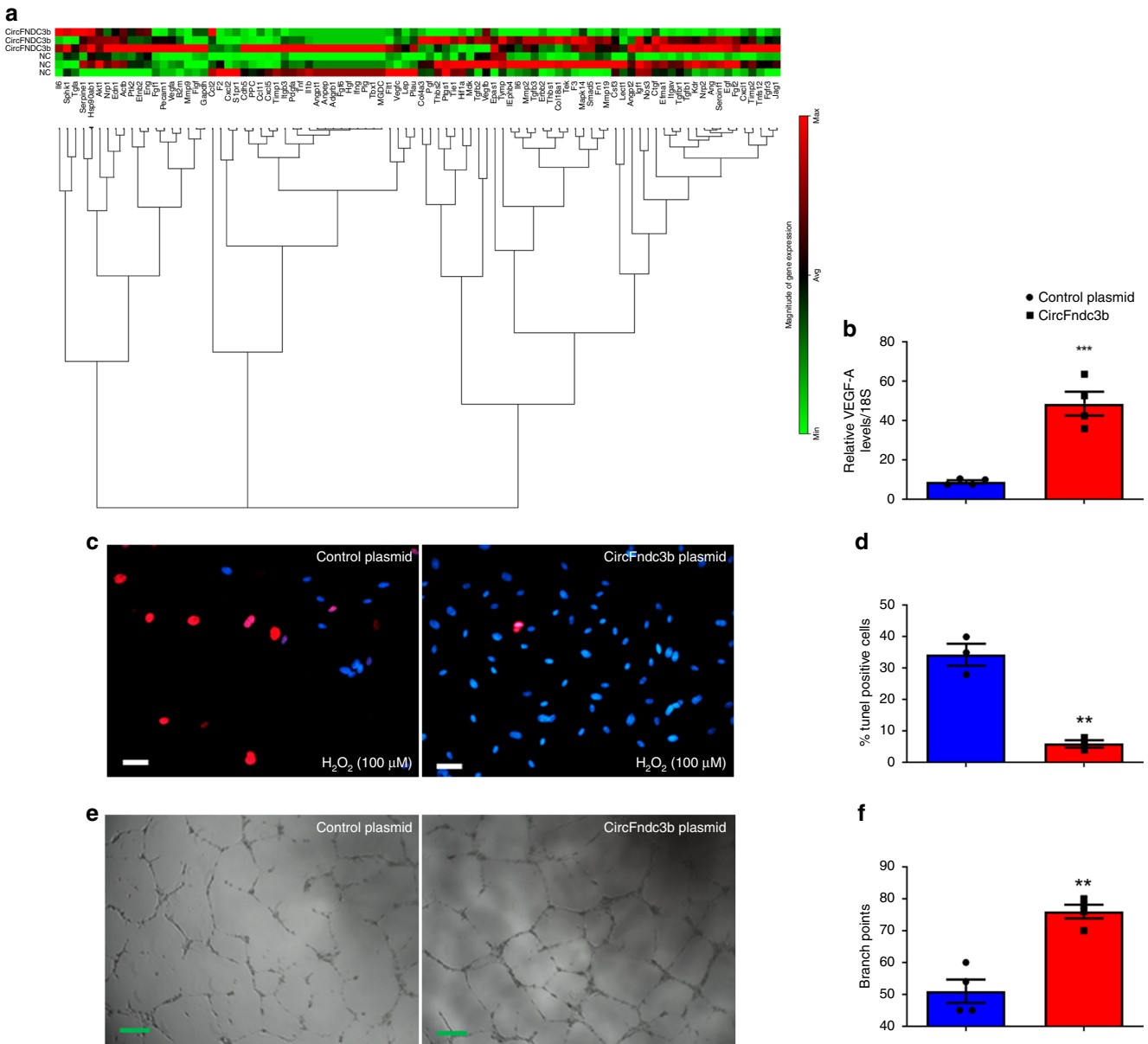

**Fig. 3** Overexpression of circFndc3b enhances endothelial cell function in vitro: **a** Heat map showing differentially expressed genes in control plasmid or circFndc3b overexpression plasmid-treated MCECs based on RT-PCR angiogenesis array. $N = 3$ independent experiments/group; **b** RT-qPCR validation of VEGF-A expression in control plasmid and circFndc3b overexpression plasmid-treated MCECs, data normalized to 18S rRNA. Data are Mean ± SEM of four independent experiments/group. ***$p < 0.001$ vs plasmid control (two-sided unpaired students t-test). **c**, **d** Representative photomicrographs (100 μm) of TUNEL staining in control plasmid or circFndc3b overexpression plasmid-treated endothelial cells. Quantification of TUNEL + cells presented as the % TUNEL-positive cells and DAPI-stained nuclei. Data are Mean ± SEM of three independent experiments/group. **$p < 0.01$ vs plasmid control (two-sided unpaired students t-test); **e** Representative photomicrographs (200 μm) of tube formation in control or circFndc3b overexpression plasmid-treated HUVECs; **f** Quantification of branch points. Data are Mean ± SEM of four independent experiments/group. **$p < 0.01$ vs control plasmid (two-sided unpaired students t-test)

control or AAV9 circFndc3b at 1, 2, 4, and 8 weeks after induction of MI (Supplementary Table 3). Percent Ejection fraction (EF) and fractional shortening (FS) was similar at baseline in all the groups. Strikingly, compared to AAV9 control and saline groups, administration of AAV9 circFndc3b significantly improved %EF (Fig. 4f) and %FS (Fig. 4g) at 2, 4, and 8 weeks after MI. Analysis of LV internal diameter during systole (LVIDs) and diastole (LVIDd) revealed a significant restoration of LV dimension with AAV9 circFndc3b treatment (Fig. 4h, i). As circRNAs are very stable molecules, we performed toxicological studies to evaluate possible adverse effects of AAV9-mediated gene delivery of circFndc3b. No significant changes were observed when comparing plasma samples collected from MI + AAV9 CircFndc3b, MI + AAV9 control or sham surgery (8 weeks post MI). Clinical chemistry parameters including Interleukin 6 (IL-6; a proinflammatory marker) (Supplementary Fig. 5A), aspartate aminotransferase (AST; liver toxicity marker) (Supplementary Fig. 5B), and cholinesterase (ChE; neurotoxicity marker) (Supplementary Fig. 5C) were all unchanged among the groups. These data suggest that exogenous gene delivery of

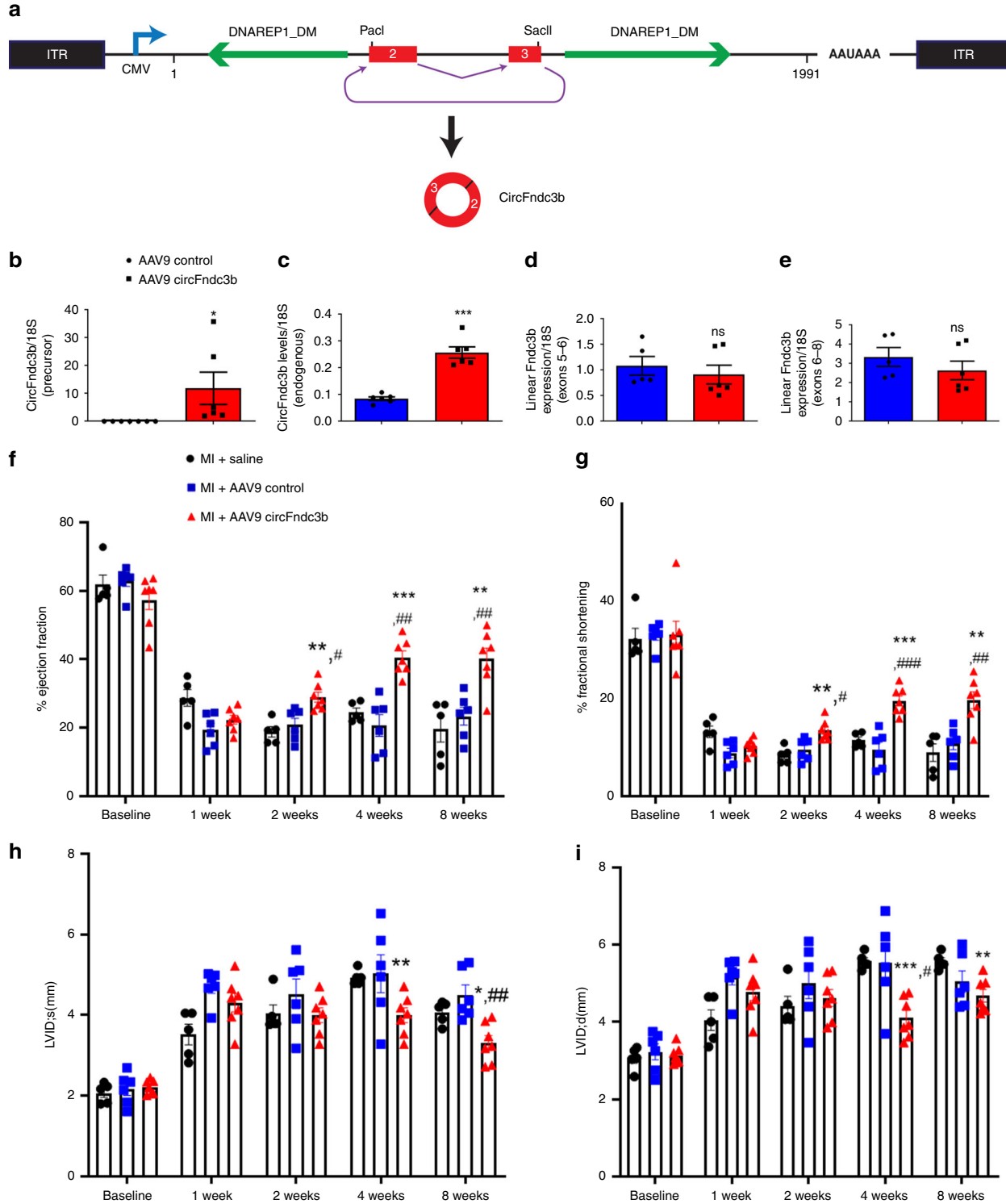

**Fig. 4** AAV9 mediated overexpression of circFndc3b improves cardiac function 8 weeks after MI in mice: **a** AAV9 vector expressing circFndc3b. **b, c** RT-qPCR analysis of precursor-circFNDC3b ($n = 6$ mice/group) and endogenous circFndc3b expression ($n = 6$ mice/group) in AAV9 circFndc3b or AAV9 control treated hearts at 8 weeks post MI. $n = 5$–7/group. *$p < 0.05$, ***$p < 0.001$ vs AAV9 control (two-sided unpaired students t-test); **c, d** RT-qPCR analysis of linear Fndc3b mRNA expression in AAV9 circFndc3b ($n = 5$ mice/group) or AAV9 control treated hearts ($n = 6$ mice/group) at 8 weeks post MI. Not significant (ns) vs AAV9 control (two-sided unpaired students t-test); **f–i** AAV9 mediated overexpression of circFndc3b ($n = 7$ mice) improved LV function (% ejection fraction, % fractional shortening, LVID; s and LVID; d measured by echocardiography compared to AAV9 control ($n = 6$ mice) or saline treatment groups ($n = 5$ mice). Data are Mean ± SEM. *$p < 0.05$, **$p < 0.01$, ***$p < 0.001$ vs AAV9 control treated hearts. #$p < 0.05$, ##$p < 0.01$, ###$p < 0.001$ vs saline treated hearts, non-significant between saline and AAV9 control treated hearts (Two-way ANOVA). The round dots represent AAV9 control and square denotes AAV9-circFndc3b

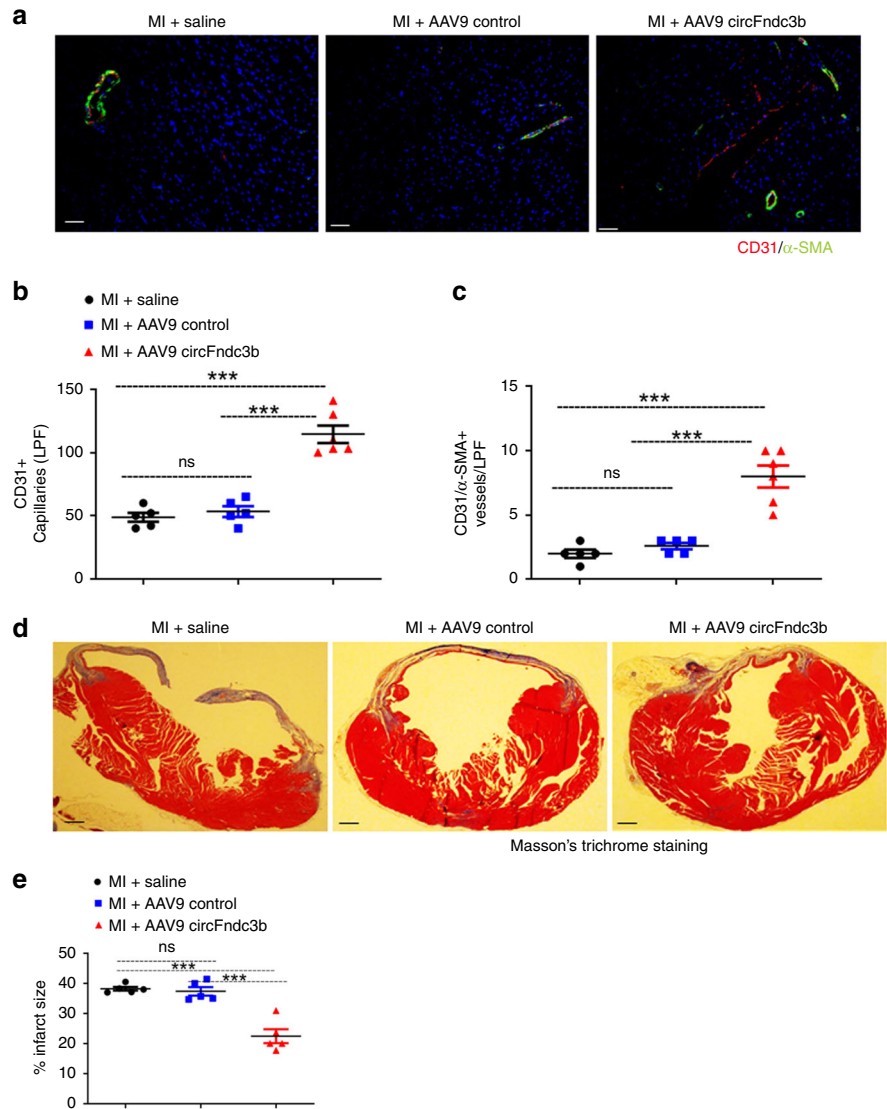

**Fig. 5** AAV9 mediated overexpression of circFndc3b enhances neovascularization and reduces fibrosis after MI in mice: **a** Representative images of capillary density and arterioles measured in border zone of LV infarct at 8 weeks post-MI in AAV9 control or AAV9 circFndc3b or saline treated hearts. Capillaries were stained with CD31 + (red) and arterioles stained with α-SMA (green) and nuclei were counterstained with DAPI (blue). (Scale bar 100 μm); **b**, **c** Quantification of border zone capillary number and arterioles across treatments presented as the number of CD31 + positive capillaries, α-SMA + arterioles and DAPI-stained nuclei per low power field (LPF). $n = 5$ mice/each group. ***$p < 0.001$ vs AAV9 control or saline treated hearts (one-way ANOVA); (**d** Representative images of Masson's trichrome stained heart sections in the saline or AAV9 control or AAV9 circFndc3b treated hearts at 8 weeks post-MI; **e** Quantitative analysis of infarct size at 8 weeks post-MI in saline or AAV9 control or AAV9 circFndc3b treated hearts; $n = 5$ mice/each group. ***$p < 0.001$ vs AAV9 control or saline treated hearts (one-way ANOVA)

circFndc3b is safe and attenuates MI-induced cardiac dysfunction.

**AAV9-mediated overexpression of circFndc3b enhances neo-vascularization and reduces fibrosis after MI**. Next, we determined the effect of circFndc3b overexpression on myocardial neovascularization. We found increased capillary density and α-SMA⁺ arterioles in the border zone of the infarcted hearts at 8 weeks post-MI in mice that received AAV9 circFndc3b compared to those receiving AAV9 control or saline (Fig. 5a–c). To determine whether improved capillary density after MI may improve cardiac remodeling, we assessed infarct size by Masson's trichrome stain at 8 weeks post-MI in mice. Overexpression of circFndc3b resulted in a significant reduction in the infarct size compared to AAV9 control or saline groups (Fig. 5d, e).

**miRNA binding sites are not critical for circFndc3b function**. Emerging evidence suggests that some circRNAs act as micro-RNA sponges[31,34]. CircFndc3b is predicted to potentially bind miR-93-3p, miR-412-3p, miR-298-5p, miR-7231-3p, and miR-6998-3p (Supplementary Fig. 6). To determine if any of these microRNA target sites are functional, we inserted the circFndc3b sequence into the 3′UTR of luciferase and measured luciferase expression after each miRNA mimic was co-transfected into MCECs. Compared with a scrambled control miRNA mimic, 3 of the 5 miRNA mimics (miR-298-5p, miR-412-3p, and miR-93-3p) reduced luciferase reporter activity, suggesting that circFndc3b may indeed bind these miRNAs (Fig. 6a). Of note, miR-93-3p, miR-412-3p, and miR-298-5p levels are often increased in the LV tissue of the post-MI hearts (Fig. 6b–d) and in H9c2 cells subjected to hypoxia (1% O₂) and serum deprivation for 48 h (Supplementary Fig. 7A–C). To determine whether the miR-93-

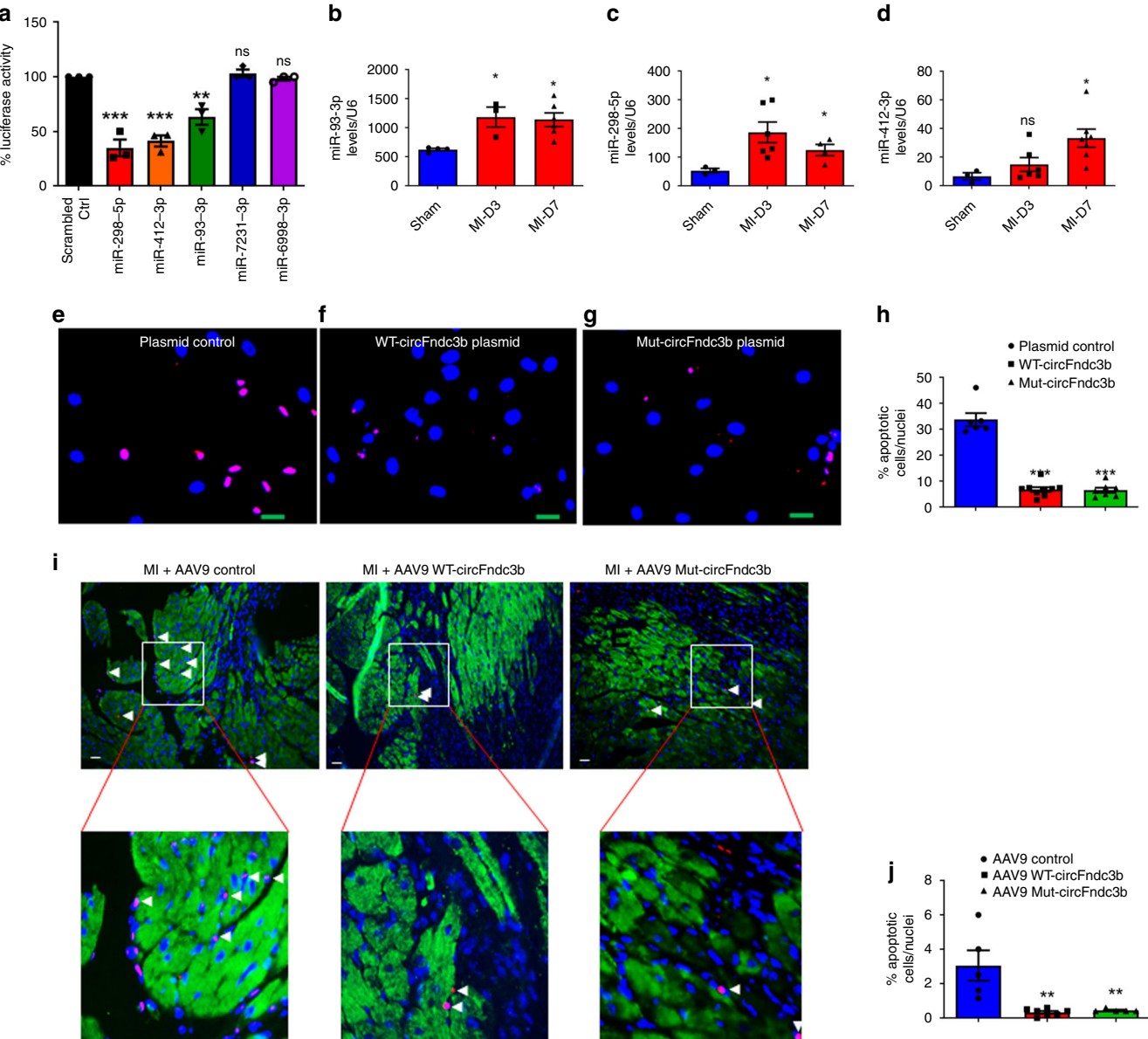

**Fig. 6** miRNA binding sites are not critical for circFndc3b function: **a** CircFndc3b sequence was cloned into the 3′UTR of the pGL3 Luciferase reporter to construct LUC-circFndc3b. MCECs were co-transfected with LUC-circFndc3b and different miRNA mimics (30 nM). Luciferase activity was detected using the dual luciferase assay at 48 h post transfection. Data are Mean ± SEM of 3 independent experiments, **$P < 0.01$ ***$p < 0.001$ vs scrambled control group (one-way ANOVA); **b–d** RT-qPCR analysis of circFndc3b target miRNAs (miR-93-3p, miR-298-5p and miR-412-3p) in 3 and 7 days post-MI hearts compared to sham, data normalized to U6 snRNA; Data are Mean ± SEM of $n = 3–5$/group. *$p < 0.05$ vs sham hearts (one-way ANOVA); **e–g** Representative images of H9c2 cells (Scale bar 20 μm) treated with control plasmid or WT-circFndc3b or Mut-circFndc3b plasmids and subjected to hypoxic stress (1% O$_2$, 48 h). TUNEL assay was performed; **h** Quantification of TUNEL + cells presented as the % TUNEL + positive cells and DAPI-stained nuclei. Data are Mean ± SEM of 6 independent experiments. ***$p < 0.001$ vs control plasmid (one-way ANOVA); **i, j** Representative images and quantification of apoptotic cardiomyocytes in the myocardium at 10 days after MI in AAV9 control or AAV9 WT-circFndc3b or Mut-circFndc3b administered mice. TUNEL staining for detecting apoptosis (red) of cardiomyocytes (α-SA, green florescence) and DAPI (blue) for nuclear staining. Arrows indicate TUNEL and α-SA + cells (Scale bar 100 μm). $n = 3$/each group. ***$p < 0.001$ vs AAV9 control (two-sided unpaired students t-test)

3p, miR-298-5p, and/or miR-412-3p binding sites are required for circFndc3b functionality, we generated a mutant circFndc3b overexpression plasmid in which all of the predicted miR-93-3p, miR-298-5p, and miR-412-3p sites were mutated (Supplementary Fig 8A). Expression of the mutant circFndc3b was higher than WT circFndc3b from the plasmids (Supplementary Fig. 8B), but neither plasmid affected linear Fndc3b mRNA levels (Supplementary Fig. 8C–D). Next, we subjected H9c2 cells to hypoxia (1% O$_2$) and serum deprivation conditions and tested whether

overexpression of wild type and mutant circFndc3b result in similar phenotypes after 48 h. TUNEL assays revealed that both WT circFndc3b and mutant circFndc3b reduced levels of apoptosis compared to a control plasmid (Fig. 6e, h), suggesting that the miRNA target sites within circFndc3b are dispensable in vitro.

Further, to test the functionality of the circFndc3b miR-93-3p, miR-298-5p, and miR-412-3p binding sites in cardiac endothelial cells, MCECs were transfected with a control plasmid or WT/ mutant circFndc3b overexpression plasmids (Supplementary

Fig. 9A–C). We then assessed the effect of WT circFndc3b and mutant circFndc3b on apoptosis in MCECs. Overexpression of either WT or mutant circFndc3b significantly reduced apoptosis in MCECs subjected to $H_2O_2$ (100 μM) exposure for 6 h (Supplementary Fig. 9D–G). Further, we assessed the role of WT and mutant circFndc3b on tube formation ability of endothelial cells and found significantly enhanced tube formation in both cases (Supplementary Fig. 9H–K). Nevertheless, neither WT nor mutant circFndc3b affected MCEC migration (Supplementary Fig. 9L). These observations indicate that circFndc3b does not act as miRNA sponge in either endothelial cells or cardiomyoblasts.

To then confirm these observations in vivo, we generated AAV9 viral particles expressing mutant circFndc3b with all three miR-93-3p, miR-298-5p and miR-412-3p interaction sites mutated. Intra-myocardial injection of $1 \times 10^{12}$ vp/ml AAV9 WT-circFndc3b or AAV9 Mut-circFndc3b yielded significantly elevated circFndc3b levels in LV tissues compared to AAV9 controls at 10 days post-MI mouse hearts. Notably, mutant circFndc3b expression was decreased compared to WT circFndc3b (Supplementary Fig. 8E–G). Upon examining cardiomyocyte apoptosis in the border zone of the infarct, we observed a larger number of apoptotic cells in the saline group compared with mice that received WT or mutant circFndc3b (Fig. 6i–j). These data suggest that circFndc3b protects against cardiomyocyte apoptosis in the ischemic myocardium, leading to enhanced myocardial repair and LV function. Importantly, these data also suggest that circFndc3b does not act as a miRNA sponge in vitro or in vivo.

The protein derived from the Fndc3b gene is well known to target PI3 kinase (PI3K) and AKT survival signaling in cancers[29,30]. Therefore, we asked if circFndc3b may also regulate the PI3K/AKT signaling pathway. However, overexpression of circFndc3b had no effect on the levels of PI3K (Supplementary Fig. 10A–B) or total and phosphorylated AKT (Supplementary Fig. 10C–D) in cardiac endothelial cells.

**A circFndc3b-FUS-VEGF-A signaling network regulates cardiac endothelial cell function.** Besides acting as miRNA sponges, circRNAs have been shown to function by interacting with RNA binding proteins[26,35,36]. To explore this possibility, we performed an in silico analysis using circInteractome[35] and found that circFndc3b has binding sites for Argonaute-2 (AGO2) and FUS (Supplementary Fig. 11A). Cell lysates were prepared from MCECs and subjected to RNA binding protein immunoprecipitation (RIP) with mouse IgG, Argonaute-2 (AGO2), or FUS antibodies (Supplementary Fig. 11B–C), followed by RNase R digestion and RT-PCR to quantify circFndc3b levels (Fig. 7a). This experiment revealed that circFndc3b was significantly enriched upon FUS pull down, but not with AGO2 (Fig. 7b). As AGO2 is a core component of the RNA-induced silencing complex (RISC)[37], these results again suggest that circFndc3b does not function as a miRNA sponge. CircFndc3b is localized in both the nucleus and the cytoplasm in MCECs (Supplementary Fig. 11D) and we investigated whether circFndc3b has an impact on FUS levels. Immunoblots revealed that overexpression of circFndc3b significantly reduced FUS levels (Fig. 7c, d). Further, VEGF-A levels were strikingly increased when either WT or mutant circFndc3b was overexpressed (Fig. 7c–e).

To investigate whether circFndc3b increases VEGF expression via FUS, we knocked down or overexpressed FUS in MCECs in the presence or absence of circFndc3b (Supplementary Fig. 11E) and then assessed VEGF levels. Consistent with the preceding experiments, circFndc3b overexpression increased VEGF expression. On the other hand, VEGF up-regulation was blunted by

FUS overexpression alone or in combination with circFndc3b overexpression (Fig. 7f, g). This suggests an interplay between circFndc3b, FUS and VEGF

To further investigate whether circFndc3b enhances endothelial cell function via FUS, we knocked down or overexpressed FUS in MCECs in the presence or absence of circFndc3b overexpression. MCECs were first subjected to $H_2O_2$ stress to induce apoptosis. CircFndc3b overexpression or FUS silencing (using siRNA) reduced the level of apoptosis observed, whereas FUS overexpression had similar effects on cell viability as controls (Fig. 7h). Notably, however, combining FUS overexpression with circFndc3b overexpression nullified the anti-apoptotic effects of circFndc3b. Upon examining MCEC migration, we determined that FUS overexpression alone or in combination with circFndc3b overexpression significantly reduced MCEC migration, whereas combining FUS silencing with circFndc3b overexpression increased migration (Fig. 7i). Collectively, these results suggest that circFndc3b-FUS signaling can enhance endothelial cell function in vitro.

Consistent with circFndc3b overexpression causing reduced FUS and increased VEGF levels in vitro, we found strikingly reduced FUS levels (Fig. 7j) and increased VEGF protein levels in the AAV9 circFndc3b treated hearts (Fig. 7k, l and Supplementary Fig. 11F) at 8 weeks post-MI compared to controls. Collectively, these results suggest that circFndc3b-FUS-VEGF signaling regulates angiogenesis in vitro and in vivo.

## Discussion

Recently, the regulatory potential of circRNAs in cardiovascular disease has garnered significant interest, especially using whole transcriptome approaches[38]. More than 1200 circRNAs were consistently detected in human, mouse, and rat cardiac tissues[39]. Analyses of cardiac tissues in a pressure overload mouse model and in heart failure patients revealed differential expression of several circRNAs, as was also the case for another study utilizing a heart failure model in mice[40]. Further, circFOXO3 has been demonstrated to improve cardiac function after doxorubicin induced cardiac injury[22]. These findings strongly support an integral role for circRNAs in cardiac homeostasis and dysfunction. In the current study, we showed that circFndc3b is significantly repressed post-MI in mice. The Fndc3b gene has been shown to be elevated in different types of cancers and exhibits a role in survival signaling[29,30], and we reveal that exogenous delivery of circFndc3b can play an important role in attenuating left ventricular dysfunction after MI in mice.

This conclusion is based on the following experimental findings: (i) circFndc3b levels are decreased in post-MI mouse hearts as well as in heart tissues from human ischemic cardiomyopathy patients; (ii) overexpression of circFndc3b enhances the endothelial cell angiogenic response and inhibits stress-induced endothelial cell apoptosis in vitro; (iii) overexpression of circFndc3b enhances angiogenesis, reduces infarct size, and ultimately attenuates LV dysfunction after MI; (iv) circFndc3b binds to its potential target FUS; and (v) mechanistically, beneficial effects of circFndc3b appear to be mediated at least in part through a FUS/VEGF signaling axis.

We observed that circFndc3b overexpression significantly improved LV function during 8 weeks observation window, post-MI. However, maintenance of improved cardiac function for time period beyond 8 weeks remains to be determined. Furthermore, it should be noted that other circular RNAs are likely to also be key players in mediating myocardial injury and repair, as our microarray data were performed on a relatively low sample size and not all currently annotated circRNAs are present on the microarray chip. Future studies using deep sequencing

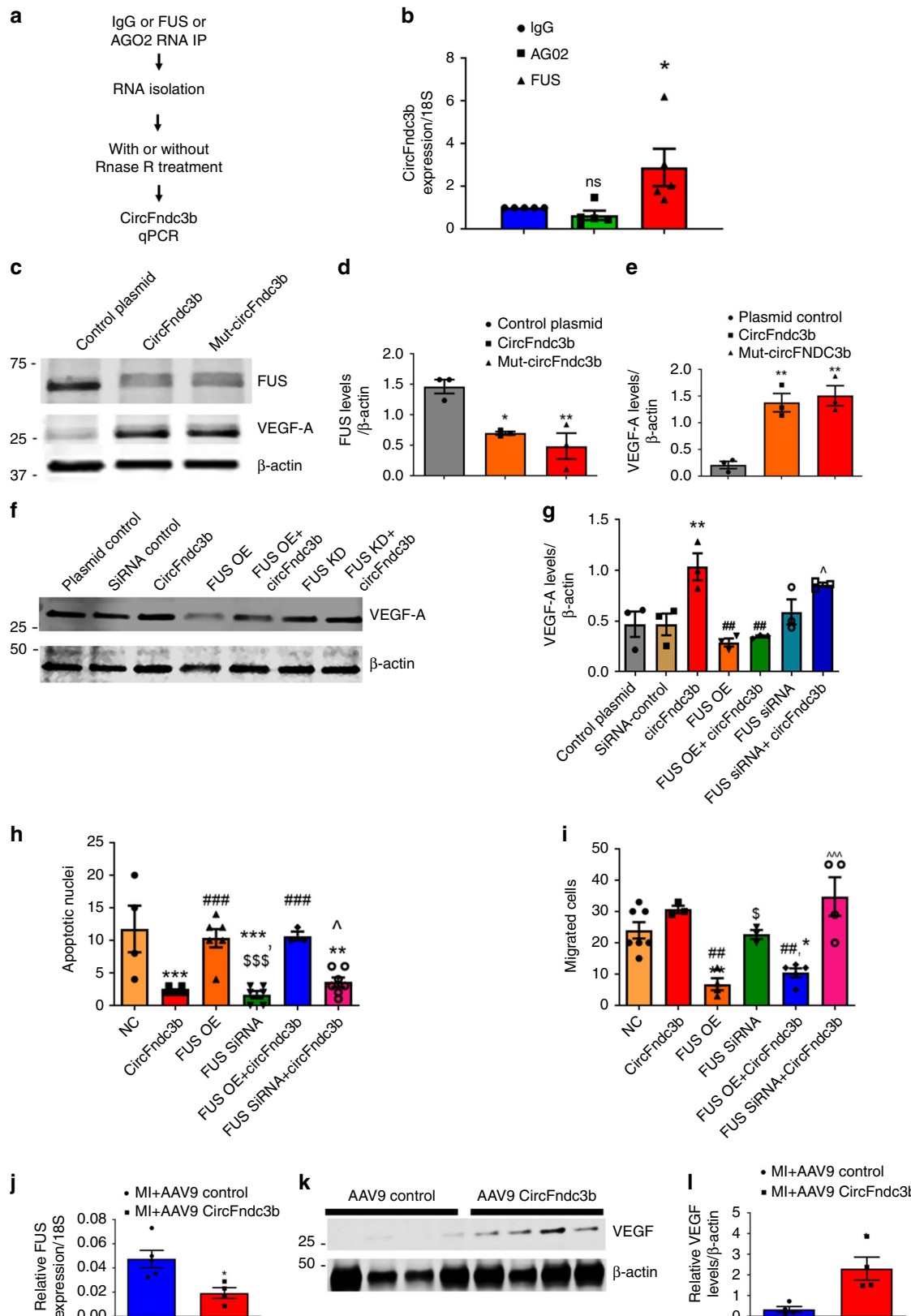

approaches will be important to delineate novel circRNAs and characterize their functional effects.

It is imperative to understand how MI alters circRNA expression and to identify the mediators that dictate function/dysfunction. Emerging reports suggest that some circRNAs have microRNA binding sites and thus can sequester homologous microRNAs using

these sites[34,40]. However, other studies indicate that very few circRNAs exhibit microRNA sponge activity[25,41,42], and we likewise find that circFndc3b does not appear to act as a microRNA sponge. Besides acting as a microRNA sponge, recent reports also suggest circRNAs can bind RNA binding proteins (RBPs)[22] and influence post-transcriptional fates (e.g., stability or translation) of

**Fig. 7** CircFndc3b-FUS-VEGF-A signaling network regulates cardiac endothelial cell function: **a** RNA binding protein immunoprecipitation analysis was carried out using anti-FUS or AGO2 or IgG control antibodies. Total RNA was isolated and digested with or without RNase R, and **b** circFndc3b levels in the samples was quantified using RT-qPCR, normalized to 18S rRNA. $n = 5$ independent experiments per group. Data were Mean ± SEM.*$p < 0.05$, **$p < 0.01$ vs IgG (one-way ANOVA); **c** Representative western blot images of FUS and VEGF-A protein levels in MCECs treated with control or circFndc3b overexpression plasmids; **d**, **e** Quantification of FUS and VEGF-A protein levels normalized to β-actin. Data were Mean ± SEM of three independent experiments. *$p < 0.05$ **$p < 0.01$ vs control plasmid (one-way ANOVA); **f** MCECs were transfected alone or in combination with FUS siRNA, FUS overexpression plasmid, or circFndc3b overexpression plasmid or their respective controls for 48 h. **f** Representative western blot images of VEGF-A; **g** Quantification of VEGF-A levels normalized to β-actin. Data shown are Mean ± SEM of 3 independent experiments **$p < 0.05$ vs control plasmid, ##$p < 0.01$ vs circFndc3b overexpressing cells. ^$p < 0.05$ vs FUS OE + circFndc3b OE treated cells (one-way ANOVA). **h** Quantification of TUNEL + MCECs presented as the % TUNEL + positive cells and DAPI-stained nuclei MCECs subjected to $H_2O_2$ (100 μM) stress. Data shown are Mean ± SEM of $n = 3–7$/group. **$p < 0.01$ ***$p < 0.001$ vs control plasmid, ###$p < 0.001$ vs circFndc3b treated cells, $$$p < 0.001$ vs FUS OE treated cells, ^$p < 0.05$ vs FUS OE + circFndc3b treated cells (one-way ANOVA). **i** MCECs were treated as above and trans-well migration assay was performed. Quantification of migrated cells presented as migrated cells counted by DAPI-stained nuclei. Data shown are Mean ± SEM of $n = 3–7$ independent experiments. *$p < 0.05$ vs control plasmid, ##$p < 0.01$ vs circFndc3b treated cells, $$p < 0.05$ vs FUS OE treated cells, ^$p < 0.05$ ^^^$p < 0.001$ vs FUS OE + circFndc3b treated cells (one-way ANOVA). **j** RT-qPCR of FUS expression in LV tissue of AAV9 control ($n = 4$ mice) or AAV9 circFndc3b ($n = 5$ mice) treated mouse hearts at 8 weeks post-MI. Data were normalized using 18S rRNA. Data shown are Mean ± SEM. *$p < 0.05$ vs AAV9 control or saline treated hearts (two-sided unpaired students $t$-test); **k**, **l** Representative western blot images of VEGF-A and its quantification in LV tissue of AAV9 control or AAV9 circFndc3b treated hearts at 8 weeks post-MI. Data was normalized using β-actin. $n = 4$ mice/group. *$p < 0.05$ vs AAV9 control or saline treated hearts (two-sided unpaired students t-test)

mRNAs[23,43,44]. In corroboration with previous reports, we find that circFndc3b interacts with FUS, an RNA binding protein that acts as a tumor suppressor gene in many human cancers[45]. FUS has been shown to play a vital role in various cellular processes, including transcription, cell cycle progression, angiogenesis, and apoptosis[45–48]. In corroboration with a recent study suggesting circular RNAs can sequester RNA binding proteins, our results also support circFndc3b interacts with FUS and regulates its levels. Reduction in FUS levels could be due to less efficient FUS biogenesis or decreased FUS mRNA stability. We speculate the latter possibility that circFndc3b stabilizes FUS mRNA stability because circFndc3b is majorly enriched in the cytoplasm. Nonetheless we cannot exclude the possibility that smaller amounts of circFndc3b we observed present in the nucleus might affect FUS biogenesis. Thus, future studies are necessary to investigate the FUS regulation by circFndc3b.

Furthermore, recent studies demonstrated exogenous FUS gene delivery significantly inhibits tumor growth[49] by activating APAF-1 induced apoptosis and inhibiting angiogenesis by reducing VEGF-A expression[46]. VEGF is the primary pro-angiogenic target for therapeutic angiogenesis[50]. Moreover, cardiac-specific deletion of VEGF-A in mice resulted in thin walled, dilated, and hypovascular hearts with basal contractile dysfunction[51]. Thus, increased VEGF expression in AAV9-circFndc3b treated mouse hearts can be associated with enhanced angiogenesis and cardiomyocyte survival leading to the improved cardiac function. Furthermore, modulation of FUS levels in our study revealed a crucial role of circFndc3b/FUS regulation of VEGF levels, suggesting that circFndc3b is a molecular regulator of cardiac function via FUS/VEGF signaling.

In summary, our data demonstrate that overexpression of circFndc3b exhibits cardio protective effects by reducing cardiomyocyte apoptosis, enhancing neovascularization, limiting infarct size, and preserving post-MI cardiac function and integrity in part through FUS/VEGF signaling mechanisms (Fig. 8). To the best of our knowledge, we show that overexpression of circFndc3b is a feasible approach to limit ischemic injury and may be an attractive option to treat patients with MI.

## Methods
**Animal model**. This study conforms to the Guide for the Care and Use of Laboratory Animals published by the US National Institutes of Health. All experiments conform to the protocols approved by the Institutional Animal Care and Use Committee of Temple University School of Medicine. Eight-weeks-old Wild-type (WT) male mice of C57BL/6 J background were procured from Jackson Research Laboratory (Bar Harbor, ME).

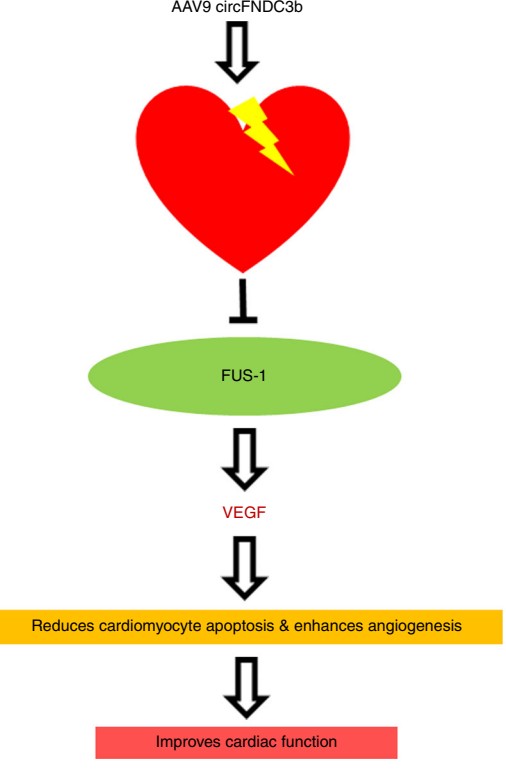

**Fig. 8** Proposed mechanism of circFndc3b mediated cardiac repair: AAV9 circFndc3b therapy increases circFndc3b levels in the post-MI heart leading to its interaction with FUS, which impacts VEGF-A signaling. This enhances angiogenesis, limits infarct size, and preserves post-MI cardiac function and integrity mediated cardiac repair. FUS, Fused in sarcoma; VEGF-A, Vascular endothelial growth factor

**Human heart tissues samples**. Heart tissue samples (n = 7) obtained from ischemic cardiomyopathy patients at the time of transplantation and were stored at tissue bio-bank at the Temple University Cardiovascular Research Center, Philadelphia, Pennsylvania and immediately frozen in liquid nitrogen and stored at −80 °C until use. Normal heart (n = 4) tissues were obtained from donor hearts not used for transplantation and were collected and stored in the same manner. Heart tissue samples non-identifying subjects' details are provided in Supplementary Table 1. The study was conducted in accordance with the Declaration of Helsinki. All tissues collection was done following protocol approved by the Temple University Institutional Review Board.

**Circular RNA microarrays**. CircRNA expression analysis was performed on total RNA extracted from the border zone of sham or MI mouse hearts (n = 2), respectively. Briefly, total RNA was digested with RNase R (Epicentre, Madison, WI, USA) to remove linear RNA and enrich for circular RNAs. The remaining RNAs were amplified and transcribed into fluorescent cRNA utilizing a random priming method (Arraystar Super RNA Labeling Kit; Arraystar) and hybridized onto the Arraystar circRNA Array (8 × 15 K, Arraystar) (Rockville, MD, USA). After washing, the arrays were scanned by the Agilent Scanner G2505C. Agilent Feature Extraction software (version 11.0.1.1) was used to analyze acquired array images. The raw expression intensities were log2 transformed and normalized by Quantile normalization. Differential analysis between groups was performed by t-test. The cutoffs were p ≤ 0.05 and |FC| ≥ 2.0. Normality was assumed for log2 transformed normalized intensity values across samples per gene. > 90% genes in our dataset passed the Shapiro–Wilk normality test.

**CircRNA-miRNA interaction**. Interactions between circRNAs and microRNAs were predicted using Arraystar's proprietary miRNA target prediction software that is based on MiRanda and TargetScan[52].

**CircFndc3b expression plasmid and AAV9 construction**. To generate a circFndc3b expression plasmid, the previously described pcDNA3.1(+) Laccase2 MCS Exon Plasmid (www.addgene.org/69893) was used[53]. This plasmid does not efficiently generate single-exon circular RNAs < 300-nt in length so a miniature intron derived from the mouse Jmjd1c gene was inserted between the two FNDC3b exons. In total, the following sequence was inserted between the PacI and SacII sites of pcDNA3.1(+) Laccase2 MCS Exon Plasmid (exons in uppercase, intron in lowercase):

GTAGCAGGTTGAGGGCAATTCTCCATGAATGTACGTCACCATGATGA TGACCGACCAGATCCCCCTGGAATTGCCACCTCTGCTCAATGGGGAGG TGGCCATGATGCCCCACTTGGTGAACGGAGAAGCAGCTCAGCAGgtaagat ttgttcctgtttttattaattgttctcctttgtttgtctaagatgagtgagcactgaatggaagttagatgtgtttggcaccggtgc atgatacagtataatttccagcttattaattatggttgtgtttaattattttttaatgaataaaaagtattctatggttttaataaa aacgttaggttataccattatgacgaccttattgaaccattatattgccatgtaccctttgtataacactttgaggatacaacc agaaaataaccattagaactttttctcttttttgacagGTCATTCTTGTCCAAGTGAACCCAGGTG AGACCTTTACCATCAGAGCCGAGGATGGGCACTTCAGTGCATCCAAG

To express a version of circFndc3b with mutated microRNA binding sites, the following sequence was similarly inserted between the PacI and SacII sites of pcDNA3.1(+) Laccase2 MCS Exon Plasmid (mutations marked in red):

GTAGCAGGTTGAGGGCAATTCTCCATGAATGTACGTCACCATGATGA TGACCGACCAGATCCCCCTGGAATTGCCACgagacgTCAATGGGGAGGT GGCCATGATGCCCCACTTGGTGAACGGAGAAGCAGCTgtcCAGgtaagatttg ttcctgtttttattaattgttctcctttgtttgtctaagatgagtgagcactgaatggaagttagatgtgtttggcaccggtgcatg atacagtataatttccagcttattaattatggttgtgtttaattattttttaatgaataaaaagtattctatggttttaataaaa acgttaggttataccattatgacgaccttattgaaccattatattgccatgtaccctttgtataacactttgaggatacaacca gaaaataaccattagaactttttctcttttttgacagGTCATTCTTGTCCAAGTGAACCCtccactGA CCTTTACCATCAGAGCCGAGGATGGGCACTTCAGTGCATCCAAG

In control experiments, the pcDNA3.1(+) Laccase2 MCS Exon plasmid was used.

For in vivo expression, the circFndc3b exonic regions (separated by the miniature Jmjd1c intron) along with the flanking Laccase2 introns that drive the backsplicing reaction were cloned into a self-complementary AAV backbone plasmid (pTRUFR)[54]. Expression of the circRNA is driven by a CMV promoter and flanked by AAV9 ITRs. As a control, the equivalent regions of the pcDNA3.1 (+) Laccase2 MCS Exon plasmid were cloned into pTRUFR. All viruses were produced using the triple transfection technique[55]. In brief, HEK293 cells were plated at 1 × 10^7 cells/15 cm plate one day prior to transfection and polyethyleneimine (Linear PEI, MW 25 kDa, Sigma-Aldrich) was used as the transfection reagent. For each 15 cm plate, the following reagents were added: 12 µg XX6-80[56], 10 µg pXR9,[57] and 6 µg pTRUFR WT-circFndc3b or pTRUFR Mut-circFndc3b. These plasmids were combined in 500 µl of DMEM without serum or antibiotics and 100 µl of PEI reagent (1 mg/ml pH5.0) was added, mixed and set aside for 5–10 min. Without changing the media on the 15 cm plates, the DNA-PEI reagent was added to the cells drop-wise. Cells were then harvested 48–72 h after transfection, collected by centrifugation and the cell pellets were re-suspended in water and freeze-thawed once prior to sonication (Branson Sonifier 250, VWR Scientific). DNase I (Sigma, MO) and MgCl₂ were then added before incubating at 37 °C for 60 min. The viruses were purified by Iodixanol step gradient centrifugation[5] followed by FPLC using HiTrap Q HP anion exchange chromatography column and AKTA Pure FPLC system (GE Healthcare Life Sciences). The purified virus particles were recovered and filtered through a 0.22 µM filter and quantified by OD 260. The viruses were aliquoted and stored at −80 °C in phosphate buffered saline (PBS) containing 5% sorbitol. AAV titers were determined by real-time PCR on plasmid genomes using the SYBR Green Master Mix (Roche).

**RNase R treatment & Quantitative PCR**. Total RNA (1 µg) was isolated from cells or tissues using miRNeasy Mini Kit (Qiagen). Nuclear and cytoplasmic RNA was extracted using nuclear and cytoplasmic RNA purification kit (Fisher scientific). For RNase R treatment, 1 µg of total RNA was incubated 30 min at 37 °C with or without 2.5 U of RNase R (Epicentre Technologies, Madison, WI). Reverse transcription (RT) was then performed using random hexamers or oligo(dT) and SuperScript III (Invitrogen) and quantitative PCR (qPCR) was performed using TaqMan master or SYBR

Green master mix (Applied biosystems). To quantify expression of circRNA transcripts, divergent primers were designed to amplify across the backsplicing junction. To quantify linear Fndc3b transcripts, convergent primers were designed to amplify exonic sequences not present in the circRNA. Primer locations are diagramed in Supplementary Fig. 1 and primers sequences are listed in Supplementary Table 2. Amplification was performed using the StepOnePlus Real-Time PCR System (Applied Biosystems, Foster City, CA) and Ct thresholds were determined by the software. Expression was quantified using 2^(delta-Ct) method using 18S rRNA (for mRNA/circRNA) or U6 small nuclear RNA (for miRNA or nuclear RNA fraction) as reference genes. To confirm the backsplicing junctions of the endogenous and plasmid/AAV-derived circRNAs, divergent primers were used for PCR amplification as diagramed in Supplementary Fig. 3. PCR products were cloned into pGEM-T Easy and subjected to Sanger sequencing.

**RNA Immunoprecipitation (RIP)**. MCECs (3 × 10^7 cells) were lysed in co-IP buffer (500 µl) and incubated with 5 µg of AGO2 (Cell signaling technologies#2897), FUS, or IgG primary antibodies at 4 °C overnight. Protein A-Sepharose (50% slurry) was then added to each sample and incubated at 4 °C for 4 h. The pellets were washed 3X with PBS and re-suspended in 0.5 ml Tri Reagent (Sigma-Aldrich). The eluted co-precipitated was treated with or without RNase R and RT-qPCR for circFndc3b was performed as described above.

**Myocardial infarction and study design**. Mice were anesthetized with 2% iso-flurane inhalation with an isoflurane delivery system (Viking Medical, Medford, NJ) and were subjected to MI by ligation of left anterior descending coronary artery (LAD) as described previously[2,58]. Immediately after LAD ligation, mice received intramyocardial injection of 1 × 10^12 vp/ml AAV9 circFndc3b (n = 10) or AAV9 control (n = 10) or saline (n = 10) in a total volume of 25 µl at 5 different sites (basal anterior, mid anterior, mid lateral, apical anterior, and apical lateral) in the peri-infarct area. All the mice were followed up for LV functional changes at 1, 2, 4 and 8 weeks using echocardiography and structural remodeling at 8 weeks post-MI.

**Isolation of adult mouse cardiomyocytes**. Adult mouse cardiomyocytes were isolated as described previously[58–60] with minor modifications. Briefly, 10- to 12 week-old WT (C57BL/6 J) mice were heparinized (50 USP units) and anesthetized under light anesthesia with Avertin (160 mg/kg BW, i.p.). The heart was quickly removed from the chest, cannulated through the ascending aorta, and mounted on a Langendorff perfusion apparatus. The heart was then retrogradely perfused (3 ml/min) for 5 min at 37 °C with constant pressure using pre-filtered Ca²⁺-free bicarbonate-based buffer containing 120 mM NaCl, 5.4 mM KCl, 1.2 mM MgSO₄, 1.2 mM NaH₂PO₄, 5.6 mM glucose, 20 mM NaHCO₃, 10 mM 2,3-butanedione monoxime (BDM; Sigma), and 5 mM taurine (Sigma), gassed with 95% O₂–5% CO₂. Enzymatic tissue digestion was initiated by adding collagenase type B (0.5 mg/ml; Boehringer Manheim), collagenase type D (0.5 mg/ml; Boehringer Manheim), and protease type XIV (0.02 mg/ml; Sigma) to the perfusion solution for 3–5 min, after which 50 µM Ca²⁺ was added to the enzyme solution. Once the heart became swollen and yellowish in color, it was quickly removed from the apparatus and manually separated into small pieces for further digestion (in a shaking water bath for 10 min at 37 °C) in the same enzyme solution. The suspension containing the dispersed myocytes was filtered through 100 µm filter and transferred to stopping buffer (perfusion buffer with the addition of 10% FBS) and gently centrifuged at 500 rpm for 30 s. Cardiomyocyte Ca²⁺ reintroduction and cell purification was performed by adding the cells to progressively increased CaCl₂ buffer concentrations (125 µM, 250 µM, 500 µM) and allowed to gravity sediment in 10 ml for 10 min each.

**Cardiac endothelial cell isolation**. Mice were euthanized by an overdose of Avertin (200 mg/kg BW, i.p.). To obtain single-cell suspensions from heart tissue, LV tissue was excised, minced with a fine scissor prior to digestion in 450 U/ml collagenase I, 60 U/ml DNase I, and 60 U/ml hyaluronidase (Sigma-Aldrich) for 1 h at 37 °C under agitation (750 rpm). Cells were then triturated through a 40 µm nylon mesh (BD Falcon), washed and centrifuged (8 min, 300 g, 4 °C). Cardiac endothelial cells were isolated by magnetic bead separation using CD31 + beads and further purity was confirmed by FACS analysis of CD31⁺ (clone 390) cells using LSR-II flow cytometer.

**Echocardiography**. Mice were anesthetized with 2% isoflurane inhalation with an isoflurane delivery system (Viking Medical, Medford, NJ). Transthoracic two-dimensional M-mode echocardiogram was obtained using Vevo 770 (Visual Sonics, Toronto, Canada) equipped with 30 MHz transducer. Echocardiographic studies were performed before MI (baseline) and at 1, 2, 4 and 8 weeks post-MI on mice anesthetized with a mixture of 1.5% isoflurane and oxygen (1 L/min). The internal diameter of the LV was measured in the short-axis view from M-mode recordings; ejection fraction (EF) and fractional shortening (FS) were calculated using corresponding formulas as previously described[2,58].

**Morphometric studies**. Mice were euthanized by an overdose of Avertin (200 mg/kg BW, i.p.). The hearts were fixed by perfusion with 10% buffered formalin. Hearts were then cut into 3 slices (apex, mid-LV and base) and paraffin embedded. Morphometric analysis including infarct size, wall thickness and percent fibrosis

was performed on Masson's trichrome stained tissue sections using Image-J software (NIH, version 1.30, http://rsb.info.nih.gov/ij/). Fibrosis area was measured to determine percent fibrosis[2].

**Histology**. Immunofluorescence staining for tissue sections was performed as described previously[9]. In mice which received control AAV9 or AAV9 circFndc3b, the formation of a new capillary network was assessed by CD31 (1:100, R&D systems# AF3628) and α-SMA (1:250, Sigma Aldrich# A2547) staining as described before[2], 10 randomly selected low-power visual fields (LPF) 8 weeks post-MI. Nuclei were counter-stained with 4,6-diamidino-2-phenylindole (DAPI, 1:10,000, Sigma Aldrich, St Louis, MO) and sections were examined with a fluorescent microscope (Nikon, Japan).

**Western blot analysis**. Tissue from the LV infarct border zone or cell lysates were prepared using ice-cold radio immunoprecipitation assay buffer (RIPA; 158 mM NaCl, 10 mM Tris HCl, pH 7.2, 1 mM ethylene glycol tetra-acetic acid (EGTA), 1 mM sodium orthovanadate, 0.1% SDS, 1.0% Triton X-100, 1% Sodium deoxycholate, 1 mM phenylmethylsulfonyl fluoride). Proteins (50 μg) were electrophoresed and analyzed using anti-FUS (1:1000, Cell signaling technologies #4885 s), anti- VEGF-A (1:1000, Abcam #181300), PI3K kinase p110δ (1:200, Santa Cruz: sc-55589), AKT (1:1000), phospho AKT(Thr308) (1:1000, Cell signaling technologies #9275) and b-actin (1:1000, Cell signaling technologies #3700) antibodies. Equal protein loading in each lane was verified using antibodies against the corresponding total protein or β-actin.

Akt Antibody- Cell signaling technologies #9272.

**Endothelial cells and treatments (in vitro studies)**. The mouse cardiac endothelial cell line (MCECs) was transiently transfected with 1 μg of control or circFndcC3b overexpression plasmid for 24–48 h. Cells were harvested and changes in the expression of CircFndc3b, linear Fndc3b, FUS, or VEGF-A were measured by RT-qPCR or western blot. Results are represented as Mean ± SEM for three independent experiments.

**Transfection protocol**. For circFndc3b transfection or control or FUS overexpression plasmid or FUS siRNA, 1 μg of circFndc3b plasmid was added in 500 μl of Opti-MEM, followed by addition of 3 μl of Lipofectamine RNAiMAX in 500 μl of Opti-Mem and incubated for 5 min. The two mixtures were pooled and incubated further for 10 min at RT. The respective transfection mixture was then added to the cells and mixed by gently swirling the plate. The plate was then incubated at 37 °C for 18 h in a 5% CO2 incubator. After incubation, media was added to the cells, and the cells were further incubated until harvest i.e., 48 h from the beginning of transfection. CircFndc3b expression or target mRNA/protein expression was determined by quantitative real-time PCR (RT-qPCR) or western immunoblotting, respectively.

**Isolation of neonatal rat cardiomyocytes and treatments**. NRCM were prepared by enzymatic digestion of hearts obtained from newborn (0–2 day old) Sprague–Dawley rat pups using percoll gradient centrifugation and plated on six-well cell culture grade plates (coated with collagen IV) at a density of $0.85 \times 10^6$ cells/well in DMEM/M199 medium and maintained at 37 °C in humid air with 5% $CO_2$. Cells were treated with plasmid control or circFndc3b overexpression plasmid and subjected to 100 μM $H_2O_2$ stress and TUNEL assay was performed.

**$RT^2$ Profiler PCR Array**. The expression of 84 angiogenic genes was evaluated using a mouse $RT^2 Profiler^{TM}$ PCR Array (Qiagen). MCECs were transfected with circFndc3b overexpression plasmid or control plasmid for 48 h. After incubation, the cells were lysed directly in TRIzol reagent (Invitrogen) for subsequent RNA isolation. cDNA preparation and qPCR was performed as per the manufacturer's instructions.

**Tube formation assay**. Human umbilical vein endothelial cells (HUVECs, $1.5 \times 10^4$ cells) were transfected with circFndc3b overexpression plasmid or control plasmid for 48 h and plated on 120 μl Matrigel (BD Falcon) in a 48-well plate. After incubation at 37 °C in an atmosphere of 5% $CO_2$ for 16 h, gels were observed by using a phase contrast microscope (×4) (Nikon TS100). The branch points for each tube structure were counted in each image.

**Apoptosis Assay**. MCECs or H9c2 cells or AC-16 cells were treated with control plasmid or circFndc3b overexpression plasmid for 24 h. Thereafter, cells were subjected to $H_2O_2$ insult (100 μm) or hypoxia and serum starvation for 18 h and cells were evaluated for apoptosis by TUNEL staining. Apoptosis was measured with the TMR cell death detection kit (Roche Diagnostics) following the manufacturer's instructions.

**Luciferase assay**. MCECs were cultured at 5% $CO_2$, 37 °C (Lonza EGM-2 Bullet Kit CC-3162). MCECs were co-transfected with miRNA mimic miR-93-3p or miR-298-5p or miR-412-3p or miR-7231-3p or miR-6998-3p and corresponding controls (30 nM) (Applied Biosystems) and a reporter plasmid containing the 3′ UTR of circFndc3b

inserted downstream of the luciferase reporter gene (pEZX-circFndc3b-UTR; GeneCopoeia; Rockville, MD) using Lipofectamine 2000 (Invitrogen) in a 48-well plate. Twenty-four hours after transfection, a luciferase assay was performed on cell-culture supernatant using Secrete-Pair dual luminescence kit (GeneCopoeia; LabOmics).

**Statistics**. Data are expressed as Mean ± SEM. Analyses were performed using Prism 8.1.2 (GraphPad Software Inc.). Mean of the groups were compared using a student t-test (for 2 groups) and ANOVA, followed by Bonferroni post-tests (for >2 groups). P values of <0.05 indicate statistical significance.

**Reporting Summary**. Further information on research design is available in the Nature Research Reporting Summary linked to this article.

## Data availability
The data that support the findings of this study are available from the corresponding author upon reasonable request. The microarray data that support the findings of this study are available at Gene Expression Omnibus (GEO), with the accession number GSE133503. The source data underlying Figs. 1B–E, G, 2B–E, 3B, D, G, 4B–I, 5B–C, 6A–C, H, J, 7B–L. Supplementary figs. 2A–D, 4A, C, D, 4F–I, 5A–C, 7A–C, 8C–G, 9A–C, 9G, 9K–L, 10A–D, 11B–D, 11F are provided as source data file.

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

## Acknowledgements

This work was partially supported by National Institutes of Health grants HL091983, HL126186, & HL134608 (to R.K.), S10OD021583 (to W.J.K.) . V.N.S.G. is supported by American Heart Association Career Development Award 18CDA34110277. J.E.W. is a Rita Allen Foundation Scholar and is supported by National Institutes of Health R35-GM119735.

## Author Contributions

V.N.S.G.: Conception and design, collection and/or assembly of data, data analysis and interpretation, manuscript writing; S.K.V., Z.C., M.C.: Collection and data analysis; S.K.V., Z.C., D.A.G., J.E.W., S.H., W.J.K.: Conception and design; D.L., M.T., Y.Y., G.H., C.W., C.B., Y.T., V.M., J.I., L.G., S.S., E.G. and S.R.: Collection and /or assembly of data; R.K.: conception and design, final editing and approval of manuscript.

## Additional information

**Conflict of interest:** The authors declare that they have no conflict of interest.

