## [Peer Review File · Nature Communications]

Reviewers' comments:

Reviewer #1 (Remarks to the Author):

In the current study, Garikipati et al. demonstrate the role of circular RNA CircFNDC3b in myocardial infarction and show that the interaction of CircFNDC3b-FUS gene and VEGFA axis could be partially responsible for the improved cardiac remodeling after myocardial infarction. Understanding the emerging roles of circular RNA in myocardial repair is an important area of research and the current study may widen the understanding of circular RNAs in physiology by demonstrating that overexpression of CircFNDC3b can reduce apoptosis, improve neovascularization and thus can enhance cardiac function after MI. Authors further mechanistically identify RNA binding protein FUS (and not AGO2) as the possible mediator of these functions possibly by modulating the VEGFA levels. The study results appear to support the hypothesis in question, but there is much confusion present in the current form of this manuscript.

The results suggest that CircFNDC3b exerts a paracrine mode of action (modulating angiogenesis, regulating apoptosis etc.) rather than influencing cardiomyocyte turnover. It would be also vital to know if authors can describe/explain whether the effect of CircFNDC3b can be sustained in vivo (beyond the 8 week duration post-MI) to really establish therapeutic efficacy in these circumstances. Another major issue is the description of making the AAV9 expression vector for the circRNA – this needs to be made much more clear and the methods are very nebulous. This reviewer had to do additional reading to understand how the circRNA would be generated from the AAV9 vector and the references provided do not adequately explain this. Instead, an alternative reference is suggested that is much more useful: RM Meganck et al., Tissue-dependent expression and translation of circular RNAs with recombinant AAV vectors in vivo; *Molecular Therapy: Nucleic Acid*, 2018.

Additionally, throughout the manuscript there exist discrepancies in the labeling of figures and instances of mismatch with the corresponding text in the results causing confusion in interpreting the results. The interpretation of some results is not sufficiently described in detail. These aspects need to be addressed critically by the authors:

1. In the abstract, authors state that “cardiac cell fractionation revealed circFNDC3b is highly enriched.....in post-MI hearts”. Need clarification as to whether the authors meant downregulation or upregulation. From the study, it appears that circFNDC3B is downregulated post-MI.
2. In Fig.1B, it is evident that circSENP6 (a SUMOylation protein) is equally downregulated as circFNDC3b after MI. However, there is no explanation given as to why the authors only focused on CircFNDC3b. This needs clarification in the text/discussion.

3. Fig. 3A gene names not visible for the reviewer to interpret the heat map.
4. In supplementary Fig. 2A-C, authors describe transfection of H9c2 cardiomyoblast cells in the result section of the manuscript text. However, the legend wrongly states it as endothelial cells. This needs to be corrected.
5. In Fig. 4 authors demonstrate that CircFNDC3b injection was done immediately after MI. This strategy ignores the variation in ejection fraction in these mice before any treatment. Did the authors have any exclusion criteria? Authors should provide the cardiac parameters of individual mice in each experimental cohort in respective time points to clarify this issue, for the reader to understand the study design.
6. Fig. 6 legend is wrong and is confusing to the reader. Legend describe panel I, which is not there in the associated figure and the figure panels in the legend does not correspond with the figure at all.
7. The expression level of endogenous circFNDC3b in mut circFNDC3b seems to be higher and may possibly statistically significant compared to the WT circFNDC3b. Authors need to state and explain the significance of this comparison if any (Supplementary Fig.6B and 7A). In addition, in supplementary Fig.6E AAV-9WT and AAV9mut shows clear differences in the endogenous CircFNDC3B levels (mutant has low circFNDC3b vs WT) but authors give no comparative statistics or explanation for this observation. However, interestingly this change in expression did not seem to affect the efficacy of reducing the apoptosis in vivo (Fig.6H)- this is not discussed.
8. Supplementary Fig 8B panels are not explained properly for the reader to comprehend. For eg; after ctrl plasmid, there are two circFNDC3b panels. Are these two different samples?
9. Manuscript text does not cite Fig 7 C, D, E. Authors are explaining a crucial result regarding the reciprocal regulation of FUS gene and VEGFA levels. However, these figures are not called out in the result section, which makes the interpretation extremely difficult. The sequences of figure panels are not cited in order in many instances in the manuscript text, which is causing confusion.

Reviewer #2 (Remarks to the Author):

This manuscript by Garikipati and colleagues addresses the role of circular RNAs in cardiovascular disease. The authors identified circFNDC3b as a modulator of cardiac repair after myocardial infarction (MI). CircFNDC3b appeared to be down-regulated in mouse heart after myocardial infarction, and was expressed by endothelial cells and cardiomyocytes. A human ortholog of circFNDC3b was also down-regulated in ischemic human hearts. When circFNDC3b was over-

expressed in cardiac endothelial cells, VEGFA was increased, apoptosis was reduced and in vitro tube formation was stimulated. In vivo, AAV9-mediated over-expression of circFNDC3b in the heart led to a reduction of cardiomyocyte apoptosis, an activation of angiogenesis, and an improvement of left ventricular (LV) function. The authors conclude that circFNDC3b may constitute a novel target to prevent post-MI remodeling.

The following comments may help improving the quality of this manuscript.

Since FNDC3b targets the PI3K/AKT pathway, does over-expression of circFNDC3b have an effect on this pathway?

More demographic and clinical details on donors of ischemic and “normal” hearts would be required.

2 samples per group for microarray analysis is very low and has a high risk of false discovery. This should be acknowledged as a study limitation.

There is no indication of how microarray data analysis was performed. Please provide details. Is 1723 the number of circRNAs that could be detected? In this case, what was the threshold used? The microarray covers 14,236 murine circRNAs.

Was data normality considered in statistical analyses? The Methods section mentions that a t-test was used for 2-group comparisons. Was the same test used for non-Gaussian data?

How were the presumed 83 differentially expressed circRNAs identified: expression value cut-off, p-value, false discovery rate? These analytical details are important considering the low number of samples which preclude reliable statistical analyses.

Please indicate the names of the circRNAs on the heat-map of Fig 1.

Can the PCR data of circFNDC3b be confirmed using divergent primers (e.g. in Fig 1C). How was 18S selected for normalization of PCR data?

Blasting of PCR primers used to detect circFNDC3b and linear FNDC3b in mouse and human revealed unexpected findings. Please provide a picture of UCSC genome browser showing the localization of FNDC3b gene in the murine and human genomes, the localization of circFNDC3b, as well as the localization of all PCR primers. It appears that reverse and forward primers for murine circFNDC3b and linear FNDC3b are inversed in Table 1. As far as human circFNDC3b is concerned, the primers appear to recognize JMJD1c gene. Please check Table 1 thoroughly.

Fig 3C-D: indicate in the legend that cells are treated with 100 μ M H₂O₂. Why in Fig3 C plasmid-treated cells do not appear purple in the merged panel ?

Fig 4D-E: please clarify whether the adenoviruses were administered at the time of MI.

Fig 5D: was baseline infarct size of comparable magnitude between the 3 groups of mice ? It may be informative to show the evolution of infarct size between baseline and 8 weeks for each mouse, in each group.

Results related to miRNA experiments may be removed from the manuscript since miRNAs do not appear to be involved in the effects of circFNDC3b.

Reviewer #3 (Remarks to the Author):

Circular RNA CircFNDC3b Modulates Cardiac Repair after Myocardial Infarction via FUS/VEGF-A axis

Venkata Naga Srikanth Garikipati et al.

This manuscript aims to demonstrate the role of circular RNA CircFNDC3b in post-myocardial infarction repairing. The authors found that circ-FNDC3b was downregulated under stress conditions such as myocardial infarction, while exogenous application of mouse circ-FNDC3b by AAV vectors could rescue cardiac function post myocardial infarction. In contrast to many other papers, the authors nicely proved that circ-FNDC3b may bind miRNAs (in somewhat artificial luciferase assay), but that this potential sponge effect through a single binding site was not critical for circ-FNDC3b

function. Together with the data from array and Immunoprecipitation assay, the authors suggest that circ-FNDC3b acts via FUS/VEGF-A regulation.

While the topic of circRNA is novel and the candidate presented here is potentially interesting, the study appears preliminary and contains several flaws:

My major concerns are:

- 1) The extent of circFNDC3b down-regulation is impressive. The huge relative change suggests that after MI the circle is beyond the limit of detection. Is this the case? What is the range of the relative Ct values in qPCR in Sham and MI groups?
- 2) The human circFNDC3b sequences which the authors provided in supplementary Table-1 are not human circFNDC3b (hsa_circ_0001361) primers. By using in-silico PCR, these primers amplified human JMJD1C. Why is this, did the authors provide wrong primer sequences or did they amplify a wrong target? Could the authors provide the correct primers and re-test the expression of human circFNDC3b (Figure 1G)?
- 3) In the same context, as the circRNA field is still young, it would be desirable to have a schematic representation of the circ/linear RNA locus including the binding sites of divergent and convergent primers used in PCR.
- 4) Could the authors provide the sequencing data of PCR products and RNase R treatment data to prove mouse/rat/human endogenous circFNDC3b has the correct back-splice site and could resist RNase R digestion compare with other linear mRNA?
- 5) To construct the exogenous mmu-circFNDC3b overexpression plasmid and AAV plasmid, the authors insert a miniature mouse Jmjd1c intron gene between the two exons of mmu-circFNDC3b to improve overexpression system. Could the authors explain why they choose mouse Jmjd1c intron gene as the insert part? The authors must provide the data that this circFNDC3b overexpression system could generate the circFNDC3b with correct back-splice site and linear spliced-exon junctions?
- 6) The mouse circFNDC3b (exogenous) primer probe sequence could not be aligned to any FNDC3b sequence. Could the authors provide the correct mouse circFNDC3b (exogenous) primer sequence and repeat the experiments in Fig2 B, supplementary Fig 2A and Fig 4A?
- 7) The heat plot in Figure 3A is not convincing. It rather suggests that the overexpression has random consequences for the presented panel of genes. For example the 3rd circRNA samples looks similar to NC-3; CircRNA-2 like NC-2... Although the authors confirm VEGF-A, the claim that the angiogenic signature changes is an overstatement in my point of view.
- 8) Same Figure; for clarity add the treatment to the figure or at least the figure legend (e.g. Fig3B 100um H2O2)

- 9) Could the authors provide the transfection protocol in details, for example which transfection reagent they used? Also, could the authors provide the transfection efficiency data of HUVEC and MCECs, such as GFP positive cells ratio?
- 10) The authors should perform the circFNDC3b functional assays in primary cardiomyocytes instead of H9C2 cell lines?
- 11) The results of MI experiments are impressive; however, I have several concerns: First, when were the AAV particles injected relative to MI? Figure 4A, why is there no circRNA expression in the control? At least the endogenous should be detected as in Fig1? Showing EF50 and FS is redundant; some contractile parameters (strain imaging) may be included. Can the authors provide a complete table of the echo parameters as supplementary figure?
- 12) Figure 4 D suggests the same extent of loss of contractile function/cells (i.e. indistinguishable EF50 in AAV control vs. AAV-circRNA). How do the authors explain the strongly reduced infarct sizes? This looks like protection from cardiomyocyte apoptosis. This would be in line with other data the authors show, but then one would expect that EF50 is also preserved early post MI!?
- 13) It is known that AAV9 in mice does not (or hardly) infect endothelial cells. How do the authors explain the strong effect on neovascularization?
- 14) What was the post MI mortality?
- 15) Fig 7B, the authors first perform the RNA immunoprecipitation of FUS and AGO2 protein and then treated the RNA sample with RNase R. How the authors could normalize the circFNDC3b expression with 18s while 18s should already be degraded by RNase R?
- 16) The authors should provide the FUS and AGO2 immunoprecipitation WB data to prove the RIP system is successful?
- 17) The authors hypothesis that circFNDC3b could interact with FUS protein. Could the authors test which subcellular compartment is circFNDC3b enriched in? According to the papers, FUS protein is mainly located in the nucleus. RNA fish is needed to further prove the circRNA localization?
- 18) Could the authors explain how circFNDC3b bind to FUS protein and also decrease the expression of FUS? Could the authors also use RNA pulldown assay to further prove the interaction between FUS and circFNDC3b?
- 19) The authors showed circFNDC3b is a highly conserved circular RNA both in mouse and human. Could the authors check whether circFNDC3b could also rescues cardiomyocytes apoptosis in human CM such as hiPSC-cardiomyocytes as a translational model?
- 20) In fig 3C and E, supplementary fig2 D and supplementary fig7 D and F, the authors miss the scale bar for the images.
- 21) Could the authors explain the method of qRT-PCR data calculation they used in detail?

Minor: The VEGF antibody given in the supplement does not exist??

Response to Reviewers- NCOMMS-18-24111

We thank all the reviewers for their suggestions, which have all been addressed and significantly improved the manuscript. We apologize for the typographical errors and other omissions in the original submission, which made multiple parts confusing. These have all been corrected.

Reviewer #1.

In the current study, Garikipati et al. demonstrate the role of circular RNA CircFNDC3b in myocardial infarction and show that the interaction of CircFNDC3b-FUS gene and VEGFA axis could be partially responsible for the improved cardiac remodeling after myocardial infarction. Understanding the emerging roles of circular RNA in myocardial repair is an important area of research and the current study may widen the understanding of circular RNAs in physiology by demonstrating that overexpression of CircFNDC3b can reduce apoptosis, improve neovascularization and thus can enhance cardiac function after MI. Authors further mechanistically identify RNA binding protein FUS (and not AGO2) as the possible mediator of these functions possibly by modulating the VEGFA levels. The study results appear to support the hypothesis in question, but there is much confusion present in the current form of this manuscript. Specific concerns:

It would be also vital to know if authors can describe/explain whether the effect of CircFNDC3b can be sustained in vivo (beyond the 8 week duration post-MI) to really establish therapeutic efficacy in these circumstances.

We thank the reviewer for their positive comments and for raising this interesting point. In the acute MI model, post-MI follow up for 8 weeks is considered a normal follow up time for physiological functions since LV remodeling is complete by this time. Most published reports using this model typically follow up for only 4-8 weeks post-MI, so we are already looking longer than many groups. We observe continued expression of exogenously delivered circFNDC3b for 8 weeks (Fig. 4B), which suggests that the improved LV function observed at 8 weeks may be sustained for additional follow up time. This, however, would require following up of animals for longer period (up to a year or longer), and those studies obviously could not be completed within the revision window of this manuscript. We agree that the Reviewer's point is certainly interesting, but one we think is beyond the scope and message of the current study.

Another major issue is the description of making the AAV9 expression vector for the circRNA – this needs to be made much more clear and the methods are very nebulous. This reviewer had to do additional reading to understand how the circRNA would be generated from the AAV9 vector and the references provided do not adequately explain this. Instead, an alternative reference is suggested that is much more useful: RM Meganck et al., Tissue-dependent expression and translation of circular RNAs with recombinant AAV vectors in vivo; Molecular Therapy: Nucleic Acid, 2018.

We apologize for the confusion about AAV vector generation. We have now included much clearer diagrams showing how the circRNA is made from the expression plasmids (Fig. 2A) and from the AAV vector genome (Fig 4A). We have updated the methods as well as included RM Meganck et al., reference.

Additionally, throughout the manuscript there exist discrepancies in the labeling of figures and instances of mismatch with the corresponding text in the results causing confusion in interpreting the results. The interpretation of some results is not sufficiently described in detail. These aspects need to be addressed critically by the authors:

We apologize for these oversights and have made necessary corrections to reflect the suggested changes in the revised manuscript.

In the abstract, authors state that “cardiac cell fractionation revealed circFNDC3b is highly enriched.....in post-MI hearts”. Need clarification as to whether the authors meant downregulation or upregulation. From the study, it appears that circFNDC3B is downregulated post-MI.

We thank the reviewer for allowing us to clarify. We meant downregulation and we have modified the text accordingly in the revised manuscript.

In Fig.1B, it is evident that circSENP6 (a SUMOylation protein) is equally downregulated as circFNDC3b after MI. However, there is no explanation given as to why the authors only focused on CircFNDC3b. This needs clarification in the text/discussion.

Thank you for raising this interesting point. We now mention in the introduction that we focused on circFNDC3b as it originates from an oncogene FNDC3b, and this made us hypothesize that circFNDC3b may play a role in angiogenesis and/or cell survival. We fully agree that other circRNAs (like circSENP6) may also participate in the injury repair response, but it is impossible to address the mechanisms of many circRNAs in a single manuscript. As the reviewer might expect, we are currently performing experiments to address the role of circSENP6 in cardiovascular biology/pathology and we hope this will be the focus of a future publication.

Fig. 3A gene names not visible for the reviewer to interpret the heat map.

We included a better resolution image reflecting gene names in the revised figure 3A.

In supplementary Fig. 2A-C, authors describe transfection of H9c2 cardiomyoblast cells in the result section of the manuscript text. However, the legend wrongly states it as endothelial cells. This needs to be corrected.

We apologize for this error. We have now corrected it to H9c2 cells in the revised manuscript.

In Fig. 4 authors demonstrate that CircFNDC3b injection was done immediately after MI. This strategy ignores the variation in ejection fraction in these mice before any treatment. Did the authors have any exclusion criteria?

We appreciate the reviewer's concern. All surgeries were performed by a dedicated surgeon with extensive experience in this model. We have found that variation in infarct size is generally minimal in this and previous studies. The intra-myocardial delivery mode necessitated the injection of AAV9-circFNDC3b immediately following ligation to avoid a second surgery, which would have resulted in significant animal loss. Moreover, all animals showing more than 35% ejection fraction on day 7 were excluded from further follow ups. As shown by the cardiac function data (Fig. 4F-I), LV functions are equivalently depressed in treated and control animals on day 7, again suggesting equivalent infarct in two groups.

Authors should provide the cardiac parameters of individual mice in each experimental cohort in respective time points to clarify this issue, for the reader to understand the study design.

We have now included cardiac parameters of individual mice in each experimental cohort at all-time points in the revised manuscript (Fig. 4F-I and supplementary Table-3).

Fig. 6 legend is wrong and is confusing to the reader. Legend describe panel I, which is not there in the associated figure and the figure panels in the legend does not correspond with the figure at all.

We apologize for the confusion, we have now corrected the figure legends to ensure that they appropriately describe the corresponding figures.

The expression level of endogenous circFNDC3b in mut circFNDC3b seems to be higher and may possibly be statistically significant compared to the WT circFNDC3b. Authors need to state and explain the significance of this comparison if any (Supplementary Fig.6B and 7A). In addition, in supplementary Fig.6E AAV-9WT and AAV9mut shows clear differences in the endogenous CircFNDC3B levels (mutant has low circFNDC3b vs WT) but authors give no comparative statistics or explanation for this observation. However, interestingly this change in expression did not seem to affect the efficacy of reducing the apoptosis in vivo (Fig.6H) - this is not discussed.

We agree and have now included the statistics as well as added several lines of corresponding text in the revised manuscript.

Supplementary Fig 8B panels are not explained properly for the reader to comprehend. For eg; after ctrl plasmid, there are two circFNDC3b panels. Are these two different samples?

We apologize for the confusion and have now explained these panels in detail in the revised manuscript.

Manuscript text does not cite Fig 7 C, D, E. Authors are explaining a crucial result regarding the

reciprocal regulation of FUS gene and VEGFA levels. However, these figures are not called out in the result section, which makes the interpretation extremely difficult. The sequences of figure panels are not cited in order in many instances in the manuscript text, which is causing confusion.

We apologize for the confusion and now explain Figure 7 in detail in the results section of revised manuscript.

Reviewer #2

This manuscript by Garikipati and colleagues addresses the role of circular RNAs in cardiovascular disease. The authors identified circFNDC3b as a modulator of cardiac repair after myocardial infarction (MI). CircFNDC3b appeared to be down-regulated in mouse heart after myocardial infarction, and was expressed by endothelial cells and cardiomyocytes. A human ortholog of circFNDC3b was also down-regulated in ischemic human hearts. When circFNDC3b was over-expressed in cardiac endothelial cells, VEGFA was increased, apoptosis was reduced and in vitro tube formation was stimulated. In vivo, AAV9-mediated over-expression of circFNDC3b in the heart led to a reduction of cardiomyocyte apoptosis, an activation of angiogenesis, and an improvement of left ventricular (LV) function. The authors conclude that circFNDC3b may constitute a novel target to prevent post-MI remodeling.

The following comments may help improving the quality of this manuscript.

Since FNDC3b targets the PI3K/AKT pathway, does over-expression of circFNDC3b have an effect on this pathway?

We thank the reviewer raising this excellent point. Our experiments revealed circFNDC3b overexpression did not affect PI3K/AKT pathway. These data are now shown in Supplementary Figure 10A-D.

More demographic and clinical details on donors of ischemic and “normal” hearts would be required.

We have now included the demographic and disease etiology for donors/patients in Supplemental Table 1. The tissues were obtained from the Temple University tissue bank and the samples were de-identified. Complete clinical profile for individual donors is not accessible to us nor it is required since the only purpose was to compare FNDC3b expression in non-diseased and diseased cardiac tissue.

2 samples per group for microarray analysis is very low and has a high risk of false discovery. This should be acknowledged as a study limitation.

We agree with the reviewer. We have now included this statement in the discussion of the revised manuscript. This is why we only focus on circFNDC3b in this manuscript as

we were able to validate the microarray data by RT-PCR analysis in multiple cell lines and tissues.

There is no indication of how microarray data analysis was performed. Please provide details. Is 1723 the number of circRNAs that could be detected? In this case, what was the threshold used? The microarray covers 14,236 murine circRNAs. Was data normality considered in statistical analyses? The Methods section mentions that a t-test was used for 2-group comparisons. Was the same test used for non-Gaussian data?

We detected expression of 1,723 of the 14,236 mouse circRNAs on the microarray. The threshold for detection was set at $|FC| \geq 2.0$. The raw intensities were log₂ transformed and normalized by Quantile normalization. Differential analysis between groups was performed by t-test. The cutoffs were $p \leq 0.05$ and $|FC| \geq 2.0$. The normality was assumed for log₂ transformed normalized intensity values across samples per gene. > 90% genes in our dataset passed Shapiro–Wilk normality test. Therefore, we considered t-test OK for this situation. We have included these details in the Methods section of the revised manuscript.

How the presumed 83 were differentially expressed circRNAs identified: expression value cut-off, p-value, false discovery rate? These analytical details are important considering the low number of samples which preclude reliable statistical analyses.

We actually identified 82 genes with $p \leq 0.05$ and $|FC| \geq 2.0$ that were differentially expressed. This included 41 up-regulated and 41 down-regulated circRNAs in the post-MI hearts compared to sham hearts (Fig. 1A).

Please indicate the names of the circRNAs on the heat-map of Fig 1.

We included a better resolution image with the circRNA names in the revised Figure 1A.

Can the PCR data of circFNDC3b be confirmed using divergent primers (e.g. in Fig 1C). How was 18S selected for normalization of PCR data?

18S rRNA is a widely used control for qRT-PCR analyses and we used it because we find its expression did not differ much across tissues, cells, and experimental treatments. 18S rRNA has been previously used to normalize circRNA expression (e.g., Meganck et al Molecular Therapy Nucleic Acids, 2018; Benoit et al, Nucleic Acids Research, 2018).

Blasting of PCR primers used to detect circFNDC3b and linear FNDC3b in mouse and human revealed unexpected findings. Please provide a picture of UCSC genome browser showing the localization of FNDC3b gene in the murine and human genomes, the localization of circFNDC3b, as well as the localization of all PCR primers. It appears that reverse and forward primers for murine circFNDC3b and linear FNDC3b are inversed in Table 1. As far as human circFNDC3b is concerned, the primers appear to recognize JMJD1c gene. Please check Table 1 thoroughly.

We thank the reviewer for pointing out these issues with the primer table, all of which have now been corrected. As requested, we now provide schematics showing the

locations of the FNDC3b primer sequences (Supplementary Fig 1). Many very large introns are present within the FNDC3b gene, which makes it very difficult to see the exact primer locations on the UCSC genome. We have thus chosen to instead show simplified schematics of the human and mouse FNDC3b loci in Supplementary Fig 1. We have further verified that all the primer sets for other genes are now correct.

Fig 3C-D: indicate in the legend that cells are treated with 100 μ M H₂O₂. Why in Fig3 C plasmid-treated cells do not appear purple in the merged panel ?

We have included better quality images in the revised manuscript and added the clarification as requested.

Fig 4D-E: please clarify whether the adenoviruses were administered at the time of MI.

We thank reviewer for this comment. We administered adenoviruses at the time of MI and included this statement in the revised manuscript.

Fig 5D: was baseline infarct size of comparable magnitude between the 3 groups of mice ? It may be informative to show the evolution of infarct size between baseline and 8 weeks for each mouse, in each group.

We have now included cardiac parameters of individual mice in each experimental cohort at all time points in the revised manuscript (Fig. 4F-I and supplementary Table-3).

Results related to miRNA experiments may be removed from the manuscript since miRNAs do not appear to be involved in the effects of circFNDC3b.

We agree that circFNDC3b does not function as a microRNA sponge and thus all these data are negative. However, there are many other published studies in the circRNA field that are claiming that circRNAs function as microRNA sponges, even when they have a single microRNA binding site. We thus feel our study is an important addition to the field as we clearly demonstrate that having one or a few miRNA binding sites in a circRNA does not necessarily mean the circRNA modulates microRNAs. These data have thus been retained in the manuscript. Reviewer #3 in fact positively commented on the fact that we included these data.

Reviewer #3:

This manuscript aims to demonstrate the role of circular RNA CircFNDC3b in post-myocardial infarction repairing. The authors found that circ-FNDC3b was downregulated under stress condition such as myocardial infarction, while exogenous application of mouse circ-FNDC3b by AAV vectors could rescue cardiac function post myocardial infarction. In contrast to many other papers, the authors nicely proved that circ-FNDC3b may bind miRNAs (in somewhat artificial luciferase assay), but that this potential sponge effect through a single binding site was not

critical for circ-FNDC3b function. Together with the data from array and Immunoprecipitation assay, the authors suggest that circ-FNDC3b acts via FUS/VEGF-A regulation. While the topic of circRNA is novel and the candidate presented here is potentially interesting, the study appears preliminary and contains several flaws:

Major concerns:

1) The extent of circFNDC3b down-regulation is impressive. The huge relative change suggests that after MI the circle is beyond the limit of detection. Is this the case? What is the range of the relative Ct values in qPCR in Sham and MI groups?

Ct values ranged between 25- 30 in the sham and MI groups.

2) The human circFNDC3b sequences which the authors provided in supplementary Table-1 are not human circFNDC3b (hsa_circ_0001361) primers. By using in-silico PCR, these primers amplified human JMJD1C. Why is this, did the authors provide wrong primer sequences or did they amplify a wrong target? Could the authors provided the correct primers and re-test the expression of human circFNDC3b (Figure 1G)?

We thank the reviewer for pointing out these issues with the primer table, all of which have now been corrected. We now provide schematics showing the locations of the FNDC3b primer sequences (Supplementary Fig 1). Many very large introns are present within the FNDC3b gene, which makes it very difficult to see the exact primer locations on the UCSC genome. We have thus chosen to instead show simplified schematics of the human and mouse FNDC3b loci in Supplementary Fig 1. We have further verified that all the primer sets for other genes are now correct.

3) In the same context, as the circRNA field is still young, it would be desirable to have a schematic representation of the circ/linear RNA locus including the binding sites of divergent and convergent primers used in PCR.

We thank the reviewer for this comment. As mentioned above, we have included a schematic representation of all the divergent and convergent primers in Supplementary Fig 1.

4) Could the authors provide the sequencing data of PCR products and RNase R treatment data to prove mouse/rat/human endogenous circFNDC3b has the correct back-splice site and could resist RNase R digestion compare with other linear mRNA?

We have now included Sanger sequencing data of the PCR products from the endogenous human and mouse circFNDC3b transcripts, which both show the expected backsplicing junction (Supplemental Fig. S3). We also confirmed that the plasmid and AAV vectors have the expected backsplicing junctions, and these data are also included in Supplemental Fig. S3. In all cases, total RNA was treated with RNase R prior to RT-PCR, thereby confirming that the cloned products are indeed RNase R resistant.

5) *To construct the exogenous mmu-circFNDC3b overexpression plasmid and AAV plasmid, the authors insert a miniature mouse Jmjd1c intron gene between the two exons of mmu-circFNDC3b to improve overexpression system. Could the authors explain why they choose mouse Jmjd1c intron gene as the insert part? The authors must provide the data that this circFNDC3b overexpression system could generate the circFNDC3b with correct back-splice site and linear spliced-exon junctions?*

It has previously been demonstrated that the overexpression plasmid with Laccase2 introns is unable to efficiently circularize single exons less than 300-nt in length, but can efficiently circularize longer sequences (Kramer et al. 2015 Genes Dev). CircFNDC3b is naturally derived from two short exons (139 and 76 nt, respectively) that are separated by an intervening intron that is ~22-kb in length. Rather than attempting to clone the large FNDC3b intron, we tested whether inclusion of much smaller “designer” introns would be sufficient to promote production of circFNDC3b from the plasmid. Through these efforts, we found that the Jmjd1c intron works the best and, hence, this intron is used in the overexpression vectors throughout this study. It should be noted that the Jmjd1c intron is efficiently removed from the mature circRNA (Supplemental Fig. S3) and thus should have no effect on circRNA function.

6) *The mouse circFNDC3b (exogenous) primer probe sequence could not be aligned to any FNDC3b sequence. Could the authors provide the correct mouse circFNDC3b (exogenous) primer sequence and repeat the experiments in Fig2 B, supplementary Fig 2A and Fig 4A?*

We apologize for the error. We have included all correct primer sets in Supplementary Table 2.

7) *The heat plot in Figure 3A is not convincing. It rather suggests that the overexpression has random consequences for the presented panel of genes. For example the 3rd circRNA samples looks similar to NC-3; CircRNA-2 like NC-2... Although the authors confirm VEGF-A, the claim that the angiogenic signature changes is an overstatement in my point of view.*

We agree with the reviewer and omitted the statement overexpression of circFNDC3b activates angiogenic signature in the results section of the revised manuscript.

8) *Same Figure; for clarity add the treatment to the figure or at least the figure legend (e.g. Fig3B 100um H2O2)*

We regret this omission. The figure legend has been edited in the revised manuscript.

9) *Could the authors provide the transfection protocol in details, for example which transfection reagent they used? Also, could the authors provide the transfection efficiency data of HUVEC and MCECs, such as GFP positive cells ratio?*

We thank the reviewer for raising this excellent point. We have included a detailed transfection protocol in the methods section of the revised manuscript. Transfection efficiency was 16% and 22% for HUVECs and MCECs, respectively.

10) *The authors should perform the circFNDC3b functional assays in primary cardiomyocytes instead of H9C2 cell lines?*

We thank the reviewer for raising this point. We performed a new set of functional assays on NRVM cells and found that overexpression of circFNDC3b reduced apoptosis in NRVM compared to plasmid controls (Supplementary Fig.4D-F). Additionally, these experiments were also repeated in human cardiomyocyte-like cell line AC16 (Supplementary Fig.4G-I)

11) *The results of MI experiments are impressive; however, I have several concerns: First, when were the AAV particles injected relative to MI? Figure 4A, why is there no circRNA expression in the control? At least the endogenous should be detected as in Fig1? Showing EF50 and FS is redundant; some contractile parameters (strain imaging) may be included. Can the authors provide a complete table of the echo parameters as supplementary figure?*

We thank the reviewer for allowing us to clarify. We injected AAV particles at the time of MI. No expression of the circFNDC3b was observed in the AAV9 control because the primers are specifically measuring circFNDC3b derived from the vector (there are unique sequences present in the vector circRNA sequence that allow it to be distinguished from the endogenous circFNDC3b). When we measure endogenous circFNDC3b levels in AAV9 control and AAV9 circFNDC3b treated MI hearts, we do indeed see endogenous circFNDC3b in the control treated animals (Fig. C). As suggested by the reviewer, we have now included measurements of LV internal diameter during diastole (LVIDd) and systole (LVIDs), which confirmed significant restoration of LV dimension with AAV9 circFNDC3b treatment (Fig. 4F and 4G). We also included a complete table of the echo parameters as Supplementary Table 3

12) *Figure 4 D suggests the same extent of loss of contractile function/cells (i.e. indistinguishable EF50 in AAV control vs. AAV-circRNA). How do the authors explain the strongly reduced infarct sizes? This looks like protection from cardiomyocyte apoptosis. This would be in line with other data the authors show, but then one would expect that EF50 is also preserved early post MI!?*

While circFNDC3b certainly limits cardiomyocyte apoptosis (thereby enabling enhanced survival), the net improvements in cardiac functions are likely the result of a combination of enhanced vascularization (increased VEGF via FUS suppression), diminished myocyte apoptosis, and resultant better structural remodeling as indicated by diminished fibrosis. Most of the early cell death after ligation is likely via necrosis and not apoptosis. The equivalent reduction of LV functions in treated and control group would suggest the better remodeling and angiogenesis as probable factors for better LV functions.

13) *It is known that AAV9 in mice does not (or hardly) infect endothelial cells. How do the authors explain the strong effect on neovascularization?*

This is an excellent point. We agree with the reviewer that AAV9 transduction of endothelial cells is generally low (but not unheard of). We believe that enhanced neovascularization may be a result of cellular cross talk between cardiomyocytes and endothelial cells *in vivo*,

e.g. through direct transduction and/or extracellular vesicles from infected cardiomyocytes to exosomes. However, experimental validation of this at this point is beyond the scope of this study.

14) *What was the post MI mortality?*

Immediate surgery associated (within 24h post MI) mortality was 20-30% in all the groups.

15) *Fig 7B, the authors first perform the RNA immunoprecipitation of FUS and AGO2 protein and then treated the RNA sample with RNase R. How the authors could normalize the circFNDC3b expression with 18s while 18s should already be degraded by RNase R?*

We apologize for the confusion. We had 2 sets of RNA samples, one treated with and one without RNase R. We measured the 18S levels in the untreated samples to normalize circFNDC3b expression.

16) *The authors should provide the FUS and AGO2 immunoprecipitation WB data to prove the RIP system is successful?*

We now include the western blot images in Supplementary Fig. 11B-C.

17) *The authors hypothesis that circFNDC3b could interact with FUS protein. Could the authors test which subcellular compartment is circFNDC3b enriched in? According to the papers, FUS protein is mainly located in the nucleus. RNA fish is needed to further prove the circRNA localization?*

Using RT-PCR of cytoplasmic and nuclear fraction of MCECs, we find that circFNDC3b is mostly enriched in the cytoplasm although it is also expressed in nuclear fractions (Supplementary Fig.11D).

18) *Could the authors explain how circFNDC3b bind to FUS protein and also decrease the expression of FUS? Could the authors also use RNA pulldown assay to further prove the interaction between FUS and circFNDC3b?*

We thank the reviewer for these excellent points. Though we do not have exact experimental evidence to first point, we think that decreased FUS level might reflect diminished pool of unbound protein. We attempted multiple RNA pull down experiments without success. We believe that while IP of entire pool of FUS allows for the detection of bound circRNA using a more sensitive RT-PCR amplification, the inability of circRNA pulldown of FUS likely reflects the circFNDC3b bound fraction of FUS is too low to be detected by western blots. In these experiments we also observed that FUS mRNA was pulled down by circFNDC3b (data not shown). We speculate that circRNA bound FUS-mRNA is not able to translate and reduced FUS level may partly be explained by this.

19) *The authors showed circFNDC3b is a highly conserved circular RNA both in mouse and*

human. Could the authors check whether circFNDC3b could also rescues cardiomyocytes apoptosis in human CM such as hPSC-cardiomyocytes as a translational model?

As a surrogate for human cardiomyocytes, we used human ventricular myocytes (AC-16 cells) and found that overexpression of circFNDC3b reduces apoptosis in these cells when they are subjected to H₂O₂ (Supplementary Fig-4A-C).

20) In fig 3C and E, supplementary fig2 D and supplementary fig7 D and F, the authors miss the scale bar for the images.

We apologize for the omission. We have now included scale bars in all the figures suggested.

21) Could the authors explain the method of qRT-PCR data calculation they used in detail?

We have included a detailed RT-qPCR protocol in the materials and methods section of the revised manuscript.

Minor: The VEGF antibody given in the supplement does not exist??

We apologize for the typographical error. We have now included the correct catalogue number.

REVIEWERS' COMMENTS:

Reviewer #1 (Remarks to the Author):

The assessment of cardiac function remains a concern to this reviewer. The response that LV remodeling in mice is completed by 8 weeks postMI is not accurate, as it in fact persists to 6 months (Fang et al., Experimental Physiology 2002 amongst many others). A longer time line of cardiac function assessment was not an unreasonable suggestion as it has been ~7 months since this reviewer first sent in the review of the initial submission. The conclusions of the paper rest on this approach definitively improving cardiac function- and as the authors used echocardiography for the assessment (MRI is gold-standard and far more expensive than echocardiography), this was not such a difficult or unreasonable experiment to perform. Most of the other concerns of the initial review are addressed but sustained improvement of LV function remains a concern.

Reviewer #2 (Remarks to the Author):

No further comments.

Reviewer #3 (Remarks to the Author):

The authors were largely responsive and most of the points raised were answered by performing additional experiments. Some minor issues remain; the following points could be improved for a better understanding of the revised manuscript:

- 1) Fig 2B and Fig 4B, the labelling of circ-Fndc3b expression (AAV) is not accurate. This gene expression is not derived from circular RNA structure of Fndc3b. It is better to label it like e.g. AAV-pre (precursor)-circ-Fndc3b.
- 2) Regarding the previous comment #17; the authors provided data showing circ-FNDC3b was mostly enriched in the cytoplasm. Could the authors discuss more about the subcellular location and relation between circ-FNDC3b and FUS protein?

Response to Reviewers

Reviewer #1 (Remarks to the Author):

The assessment of cardiac function remains a concern to this reviewer. The response that LV remodeling in mice is completed by 8 weeks post-MI is not accurate, as it in fact persists to 6 months (Fang et al., Experimental Physiology 2002 amongst many others). A longer time line of cardiac function assessment was not an unreasonable suggestion as it has been ~7 months since this reviewer first sent in the review of the initial submission. The conclusions of the paper rest on this approach definitively improving cardiac function- and as the authors used echocardiography for the assessment (MRI is gold-standard and far more expensive than echocardiography), this was not such a difficult or unreasonable experiment to perform. Most of the other concerns of the initial review are addressed but sustained improvement of LV function remains a concern.

We thank the reviewer for this thoughtful comment. We believe that reviewers point is certainly valid and important, however we think it is beyond the scope and message of this manuscript. To address this issue, we now include a statement in results/discussion stating that “we observed that circFndc3b overexpression significantly improved LV function within 8 weeks post-MI window of observation. However, long-term maintenance of improved cardiac functions beyond 8 weeks remains to be determined”.

Reviewer #3 (Remarks to the Author):

The authors were largely responsive and most of the points raised were answered by performing additional experiments. Some minor issues remain; the following points could be improved for a better understanding of the revised manuscript:

1) Fig 2B and Fig 4B, the labelling of circ-Fndc3b expression (AAV) is not accurate. This gene expression is not derived from circular RNA structure of Fndc3b. It is better to label it like e.g. AAV-pre (precursor)-circ-Fndc3b.

As suggested by the reviewer we have included precursor-circ-Fndc3b in the Fig 2B and Fig 4B of the revised manuscript.

2) Regarding the previous comment #17; the authors provided data showing circ-FND3b was mostly enriched in the cytoplasm. Could the authors discuss more about the subcellular location and relation between circ-FND3b and FUS protein?

We thank the reviewer for the comment. We have now included a new paragraph in the discussion section in the revised manuscript.

“In corroboration with a recent study suggesting circular RNAs can sequester RNA binding proteins, our results also support circFndc3b interacts with FUS and regulates its levels. Reduction in FUS levels could be due to less efficient FUS biogenesis or decreased FUS mRNA stability. We speculate the latter possibility that circFndc3b stabilizes FUS mRNA because circFndc3b is majorly enriched in the cytoplasm. Nonetheless we cannot exclude the possibility that smaller amounts of circFndc3b we observed present in the nucleus might affect FUS biogenesis. Thus, future studies are necessary to investigate the in-depth mechanism”.